# Restart Sampling for Improving Generative Processes

**Yilun Xu**[*]
MIT
ylxu@mit.edu

**Mingyang Deng**[*]
MIT
dengm@mit.edu

**Xiang Cheng**[*]
MIT
chengx@mit.edu

**Yonglong Tian**
Google Research
yonglong@google.com

**Ziming Liu**
MIT
zmliu@mit.edu

**Tommi Jaakkola**
MIT
tommi@csail.mit.edu

## Abstract

Generative processes that involve solving differential equations, such as diffusion models, frequently necessitate balancing speed and quality. ODE-based samplers are fast but plateau in performance while SDE-based samplers deliver higher sample quality at the cost of increased sampling time. We attribute this difference to sampling errors: ODE-samplers involve smaller discretization errors while stochasticity in SDE contracts accumulated errors. Based on these findings, we propose a novel sampling algorithm called *Restart* in order to better balance discretization errors and contraction. The sampling method alternates between adding substantial noise in additional forward steps and strictly following a backward ODE. Empirically, Restart sampler surpasses previous SDE and ODE samplers in both speed and accuracy. Restart not only outperforms the previous best SDE results, but also accelerates the sampling speed by 10-fold / 2-fold on CIFAR-10 / ImageNet $64\times64$. In addition, it attains significantly better sample quality than ODE samplers within comparable sampling times. Moreover, Restart better balances text-image alignment/visual quality versus diversity than previous samplers in the large-scale text-to-image Stable Diffusion model pre-trained on LAION $512\times512$. Code is available at https://github.com/Newbeeer/diffusion_restart_sampling

## 1 Introduction

Deep generative models based on differential equations, such as diffusion models and Poission flow generative models, have emerged as powerful tools for modeling high-dimensional data, from image synthesis [23, 9, 13, 27, 28] to biological data [10, 26]. These models use iterative backward processes that gradually transform a simple distribution (*e.g.*, Gaussian in diffusion models) into a complex data distribution by solving a differential equations. The associated vector fields (or drifts) driving the evolution of the differential equations are predicted by neural networks. The resulting sample quality can be often improved by enhanced simulation techniques but at the cost of longer sampling times.

Prior samplers for simulating these backward processes can be categorized into two groups: ODE-samplers whose evolution beyond the initial randomization is deterministic, and SDE-samplers where the generation trajectories are stochastic. Several works [23, 12, 13] show that these samplers demonstrate their advantages in different regimes, as depicted in Fig. 1(b). ODE solvers [22, 16, 13] result in smaller discretization errors, allowing for decent sample quality even with larger step sizes (*i.e.*, fewer number of function evaluations (NFE)). However, their generation quality plateaus rapidly. In contrast, SDE achieves better quality in the large NFE regime, albeit at the expense of increased sampling time. To better understand these differences, we theoretically analyze SDE performance: the

---

[*]Equal Contribution.

37th Conference on Neural Information Processing Systems (NeurIPS 2023).

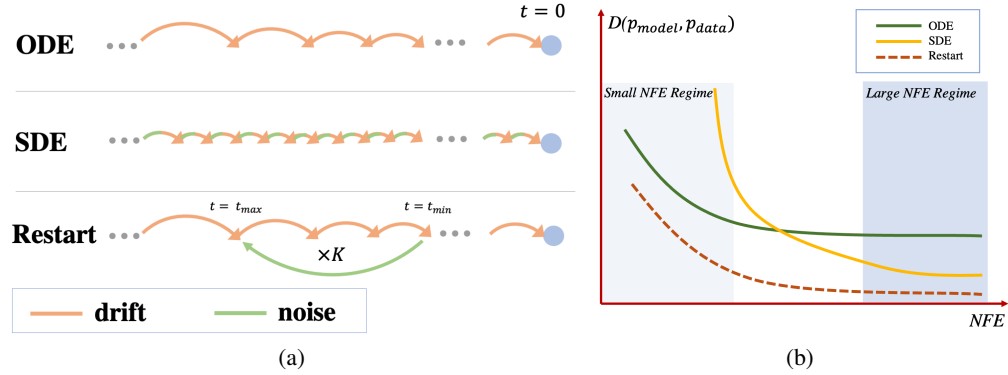

(a)                                                    (b)

Figure 1: **(a)** Illustration of the implementation of drift and noise terms in ODE, SDE, and Restart. **(b)** Sample quality versus number of function evaluations (NFE) for different approaches. ODE (Green) provides fast speeds but attains only mediocre quality, even with a large NFE. SDE (Yellow) obtains good sample quality but necessitates substantial sampling time. In contrast to ODE and SDE, which have their own winning regions, Restart (Red) achieves the best quality across all NFEs.

stochasticity in SDE contracts accumulated error, which consists of both the discretization error along the trajectories as well as the approximation error of the learned neural network relative to the ground truth drift (*e.g.*, score function in diffusion model [23]). The approximation error dominates when NFE is large (small discretization steps), explaining the SDE advantage in this regime. Intuitively, the stochastic nature of SDE helps "forget" accumulated errors from previous time steps.

Inspired by these findings, we propose a novel sampling algorithm called *Restart*, which combines the advantages of ODE and SDE. As illustrated in Fig. 1(a), the Restart sampling algorithm involves $K$ repetitions of two subroutines in a pre-defined time interval: a *Restart forward process* that adds a substantial amount of noise, akin to "restarting" the original backward process, and a *Restart backward process* that runs the backward ODE. The Restart algorithm separates the stochasticity from the drifts, and the amount of added noise in the Restart forward process is significantly larger than the small single-step noise interleaving with drifts in previous SDEs such as [23, 13], thus amplifying the contraction effect on accumulated errors. By repeating the forward-backward cycle $K$ times, the contraction effect introduced in each Restart iteration is further strengthened. The deterministic backward processes allow Restart to reduce discretization errors, thereby enabling step sizes comparable to ODE. To maximize the contraction effects in practice, we typically position the Restart interval towards the end of the simulation, where the accumulated error is larger. Additionally, we apply multiple Restart intervals to further reduce the initial errors in more challenging tasks.

Experimentally, Restart consistently surpasses previous ODE and SDE solvers in both quality and speed over a range of NFEs, datasets, and pre-trained models. Specifically, Restart accelerates the previous best-performing SDEs by $10\times$ fewer steps for the same FID score on CIFAR-10 using VP [23] ($2\times$ fewer steps on ImageNet $64 \times 64$ with EDM [13]), and outperforms ODE solvers even in the small NFE regime. When integrated into previous state-of-the-art pre-trained models, Restart further improves performance, achieving FID scores of 1.88 on unconditional CIFAR-10 with PFGM++ [28], and 1.36 on class-conditional ImageNet $64 \times 64$ with EDM. To the best of our knowledge, these are the best FID scores obtained on commonly used UNet architectures for diffusion models without additional training. We also apply Restart to the practical application of text-to-image Stable Diffusion model [19] pre-trained on LAION $512 \times 512$. Restart more effectively balances text-image alignment/visual quality (measured by CLIP/Aesthetic scores) and diversity (measured by FID score) with a varying classifier-free guidance strength, compared to previous samplers.

Our contributions can be summarized as follows: **(1)** We investigate ODE and SDE solvers and theoretically demonstrate the contraction effect of stochasticity via an upper bound on the Wasserstein distance between generated and data distributions (Sec 3); **(2)** We introduce the Restart sampling, which better harnesses the contraction effect of stochasticity while allowing for fast sampling. The sampler results in a smaller Wasserstein upper bound (Sec 4); **(3)** Our experiments are consistent with the theoretical bounds and highlight Restart's superior performance compared to previous samplers on standard benchmarks in terms of both quality and speed. Additionally, Restart improves the trade-off between key metrics on the Stable Diffusion model (Sec 5).

## 2 Background on Generative Models with Differential Equations

Many recent successful generative models have their origin in physical processes, including diffusion models [9, 23, 13] and Poisson flow generative models [27, 28]. These models involve a forward process that transforms the data distribution into a chosen smooth distribution, and a backward process that iteratively reverses the forward process. For instance, in diffusion models, the forward process is the diffusion process with no learned parameters:

$$\mathrm{d}x = \sqrt{2\dot{\sigma}(t)\sigma(t)}\mathrm{d}W_t,$$

where $\sigma(t)$ is a predefined noise schedule increasing with $t$, and $W_t \in \mathbb{R}^d$ is the standard Wiener process. For simplicity, we omit an additional scaling function for other variants of diffusion models as in EDM [13]. Under this notation, the marginal distribution at time $t$ is the convolution of data distribution $p_0 = p_{\text{data}}$ and a Gaussian kernel, *i.e.*, $p_t = p_0 * \mathcal{N}(\mathbf{0}, \sigma^2(t)\boldsymbol{I}_{d \times d})$. The prior distribution is set to $\mathcal{N}(\mathbf{0}, \sigma^2(T)\boldsymbol{I}_{d \times d})$ since $p_T$ is approximately Gaussian with a sufficiently large $T$. Sampling of diffusion models is done via a reverse-time SDE [1] or a marginally-equivalent ODE [23]:

$$\text{(SDE)} \qquad \mathrm{d}x = -2\dot{\sigma}(t)\sigma(t)\nabla_x \log p_t(x)dt + \sqrt{2\dot{\sigma}(t)\sigma(t)}\mathrm{d}W_t \qquad (1)$$

$$\text{(ODE)} \qquad \mathrm{d}x = -\dot{\sigma}(t)\sigma(t)\nabla_x \log p_t(x)dt \qquad (2)$$

where $\nabla_x \log p_t(x)$ in the drift term is the score of intermediate distribution at time $t$. W.l.o.g we set $\sigma(t) = t$ in the remaining text, as in [13]. Both processes progressively recover $p_0$ from the prior distribution $p_T$ while sharing the same time-dependent distribution $p_t$. In practice, we train a neural network $s_\theta(x, t)$ to estimate the score field $\nabla_x \log p_t(x)$ by minimizing the denoising score-matching loss [25]. We then substitute the score $\nabla_x \log p_t(x)$ with $s_\theta(x, t)$ in the drift term of above backward SDE (Eq. (1))/ODE (Eq. (2)) for sampling.

Recent work inspired by electrostatics has not only challenged but also integrated diffusion models, notably PFGM/PFGM++, enhances performance in both image and antibody generation [27, 28, 10]. They interpret data as electric charges in an augmented space, and the generative processes involve the simulations of differential equations defined by electric field lines. Similar to diffusion models, PFGMs train a neural network to approximate the electric field in the augmented space.

## 3 Explaining SDE and ODE performance regimes

To sample from the aforementioned generative models, a prevalent approach employs general-purpose numerical solvers to simulate the corresponding differential equations. This includes Euler and Heun's 2nd method [2] for ODEs (e.g., Eq. (2)), and Euler-Maruyama for SDEs (e.g., Eq. (1)). Sampling algorithms typically balance two critical metrics: (1) the quality and diversity of generated samples, often assessed via the Fréchet Inception Distance (FID) between generated distribution and data distribution [7] (lower is better), and (2) the sampling time, measured by the number of function evaluations (NFE). Generally, as the NFE decreases, the FID score tends to deteriorate across all samplers. This is attributed to the increased discretization error caused by using a larger step size in numerical solvers.

However, as illustrated in Fig. 1(b) and observed in previous works on diffusion models [23, 22, 13], the typical pattern of the quality vs time curves behaves differently between the two groups of samplers, ODE and SDE. When employing standard numerical solvers, ODE samplers attain a decent quality with limited NFEs, whereas SDE samplers struggle in the same small NFE regime. However, the performance of ODE samplers quickly reaches a plateau and fails to improve with an increase in NFE, whereas SDE samplers can achieve noticeably better sample quality in the high NFE regime. This dilemma raises an intriguing question: *Why do ODE samplers outperform SDE samplers in the small NFE regime, yet fall short in the large NFE regime?*

The first part of the question is relatively straightforward to address: given the same order of numerical solvers, simulation of ODE has significantly smaller discretization error compared to the SDE. For example, the first-order Euler method for ODE results in a local error of $O(\delta^2)$, whereas the first-order Euler-Maruyama method for SDEs yeilds a local error of $O(\delta^{\frac{3}{2}})$ (see *e.g.*, Theorem 1 of [4]), where $\delta$ denotes the step size. As $O(\delta^{\frac{3}{2}}) \gg O(\delta^2)$, ODE simulations exhibit lower sampling errors than SDEs, likely causing the better sample quality with larger step sizes in the small NFE regime.

In the large NFE regime the step size $\delta$ shrinks and discretization errors become less significant for both ODEs and SDEs. In this regime it is the *approximation error* — error arising from an

inaccurate estimation of the ground-truth vector field by the neural network $s_\theta$ — starts to dominate the sampling error. We denote the discretized ODE and SDE using the learned field $s_\theta$ as $\text{ODE}_\theta$ and $\text{SDE}_\theta$, respectively. In the following theorem, we evaluate the total errors from simulating $\text{ODE}_\theta$ and $\text{SDE}_\theta$ within the time interval $[t_{\min}, t_{\max}] \subset [0, T]$. This is done via an upper bound on the Wasserstein-1 distance between the generated and data distributions at time $t_{\min}$. We characterize the accumulated initial sampling errors up until $t_{\max}$ by total variation distances. Below we show that the inherent stochasticity of SDEs aids in contracting these initial errors at the cost of larger additional sampling error in $[t_{\min}, t_{\max}]$. Consequently, SDE results in a smaller upper bound as the step size $\delta$ nears 0 (pertaining to the high NFE regime).

**Theorem 1** (Informal). *Let $t_{max}$ be the initial noise level and $p_t$ denote the true distribution at noise level $t$. Let $p_t^{ODE_\theta}, p_t^{SDE_\theta}$ denote the distributions of simulating $ODE_\theta$, $SDE_\theta$ respectively. Assume that $\forall t \in [t_{min}, t_{max}], \|x_t\| < B/2$ for any $x_t$ in the support of $p_t$, $p_t^{ODE_\theta}$ or $p_t^{SDE_\theta}$. Then*

$$W_1(p_{t_{min}}^{ODE_\theta}, p_{t_{min}}) \leq B \cdot TV\left(p_{t_{max}}^{ODE_\theta}, p_{t_{max}}\right) + O(\delta + \epsilon_{approx}) \cdot (t_{max} - t_{min})$$

$$\underbrace{W_1(p_{t_{min}}^{SDE_\theta}, p_{t_{min}})}_{total\ error} \leq \underbrace{\left(1 - \lambda e^{-U}\right) B \cdot TV(p_{t_{max}}^{SDE_\theta}, p_{t_{max}})}_{upper\ bound\ on\ contracted\ error} + \underbrace{O(\sqrt{\delta t_{max}} + \epsilon_{approx})(t_{max} - t_{min})}_{upper\ bound\ on\ additional\ sampling\ error}$$

*In the above, $U = BL_1/t_{min} + L_1^2 t_{max}^2/t_{min}^2$, $\lambda < 1$ is a contraction factor, $L_1$ and $\epsilon_{approx}$ are uniform bounds on $\|t s_\theta(x_t, t)\|$ and the approximation error $\|t \nabla_x \log p_t(x) - t s_\theta(x, t)\|$ for all $x_t, t$, respectively. $O()$ hides polynomial dependency on various Lipschitz constants and dimension.*

We defer the formal version and proof of Theorem 1 to Appendix A.1. As shown in the theorem, the upper bound on the total error can be decomposed into upper bounds on the *contracted error* and *additional sampling error*. $TV(p_{t_{\max}}^{\text{ODE}_\theta}, p_{t_{\max}})$ and $TV(p_{t_{\max}}^{\text{SDE}_\theta}, p_{t_{\max}})$ correspond to the initial errors accumulated from both approximation and discretization errors during the simulation of the backward process, up until time $t_{\max}$. In the context of SDE, this accumulated error undergoes contraction by a factor of $1 - \lambda e^{-BL_1/t_{\min} - L_1^2 t_{\max}^2/t_{\min}^2}$ within $[t_{\min}, t_{\max}]$, due to the effect of adding noise. Essentially, the minor additive Gaussian noise in each step can drive the generated distribution and the true distribution towards each other, thereby neutralizing a portion of the initial accumulated error.

The other term related to additional sampling error includes the accumulation of discretization and approximation errors in $[t_{\min}, t_{\max}]$. Despite the fact that SDE incurs a higher discretization error than ODE ($O(\sqrt{\delta})$ versus $O(\delta)$), the contraction effect on the initial error is the dominant factor impacting the upper bound in the large NFE regime where $\delta$ is small. Consequently, the upper bound for SDE is significantly lower. This provides insight into why SDE outperforms ODE in the large NFE regime, where the influence of discretization errors diminishes and the contraction effect dominates. In light of the distinct advantages of SDE and ODE, it is natural to ask whether we can combine their strengths. Specifically, can we devise a sampling algorithm that maintains a comparable level of discretization error as ODE, while also benefiting from, or even amplifying, the contraction effects induced by the stochasticity of SDE? In the next section, we introduce a novel algorithm, termed *Restart*, designed to achieve these two goals simultaneously.

## 4 Harnessing stochasticity with Restart

In this section, we present the Restart sampling algorithm, which incorporates stochasticity during sampling while enabling fast generation. We introduce the algorithm in Sec 4.1, followed by a theoretical analysis in Sec 4.2. Our analysis shows that Restart achieves a better Wasserstein upper bound compared to those of SDE and ODE in Theorem 1 due to greater contraction effects.

### 4.1 Method

In the Restart algorithm, simulation performs a few repeated back-and-forth steps within a pre-defined time interval $[t_{\min}, t_{\max}] \subset [0, T]$, as depicted in Figure 1(a). This interval is embedded into the simulation of the original backward ODE referred to as the *main backward process*, which runs from $T$ to 0. In addition, we refer to the backward process within the Restart interval $[t_{\min}, t_{\max}]$ as the *Restart backward process*, to distinguish it from the main backward process.

Starting with samples at time $t_{\min}$, which are generated by following the main backward process, the Restart algorithm adds a large noise to transit the samples from $t_{\min}$ to $t_{\max}$ with the help of

the forward process. The forward process does not require any evaluation of the neural network $s_\theta(x, t)$, as it is generally defined by an analytical perturbation kernel capable of transporting distributions from $t_{\min}$ to $t_{\max}$. For instance, in the case of diffusion models, the perturbation kernel is $\mathcal{N}(\mathbf{0}, (\sigma(t_{\max})^2 - \sigma(t_{\min})^2)\boldsymbol{I}_{d \times d})$. The added noise in this step induces a more significant contraction compared to the small, interleaved noise in SDE. The step acts as if partially restarting the main backward process by increasing the time. Following this step, Restart simulates the backward ODE from $t_{\max}$ back to $t_{\min}$ using the neural network predictions as in regular ODE. We repeat these forward-backward steps within $[t_{\min}, t_{\max}]$ interval $K$ times in order to further derive the benefit from contraction. Specifically, the forward and backward processes in the $i^{\text{th}}$ iteration ($i \in \{0, \ldots, K-1\}$) proceed as follows:

$$\text{(Restart forward process)} \quad x_{t_{\max}}^{i+1} = x_{t_{\min}}^i + \varepsilon_{t_{\min} \to t_{\max}} \tag{3}$$

$$\text{(Restart backward process)} \quad x_{t_{\min}}^{i+1} = \text{ODE}_\theta(x_{t_{\max}}^{i+1}, t_{\max} \to t_{\min}) \tag{4}$$

where the initial $x_{t_{\min}}^0$ is obtained by simulating the ODE until $t_{\min}$: $x_{t_{\min}}^0 = \text{ODE}_\theta(x_T, T \to t_{\min})$, and the noise $\varepsilon_{t_{\min} \to t_{\max}}$ is sampled from the corresponding perturbation kernel from $t_{\min}$ to $t_{\max}$. The Restart algorithm not only adds substantial noise in the Restart forward process (Eq. (3)), but also separates the stochasticity from the ODE, leading to a greater contraction effect, which we will demonstrate theoretically in the next subsection. For example, we set $[t_{\min}, t_{\max}] = [0.05, 0.3]$ for the VP model [13] on CIFAR-10. Repetitive use of the forward noise effectively mitigates errors accumulated from the preceding simulation up until $t_{\max}$. Furthermore, the Restart algorithm does not suffer from large discretization errors as it is mainly built from following the ODE in the Restart backward process (Eq. (4)). The effect is that the Restart algorithm is able to reduce the total sampling errors even in the small NFE regime. Detailed pseudocode for the Restart sampling process can be found in Algorithm 2, Appendix B.2.

## 4.2 Analysis

We provide a theoretical analysis of the Restart algorithm under the same setting as Theorem 1. In particular, we prove the following theorem, which shows that Restart achieves a much smaller contracted error in the Wasserstein upper bound than SDE (Theorem 1), thanks to the separation of the noise from the drift, as well as the large added noise in the Restart forward process (Eq. (3)). The repetition of the Restart cycle $K$ times further leads to a enhanced reduction in the initial accumulated error. We denote the intermediate distribution in the $i^{\text{th}}$ Restart iteration, following the discretized trajectories and the learned field $s_\theta$, as $p_{t \in [t_{\min}, t_{\max}]}^{\text{Restart}_\theta(i)}$.

**Theorem 2** (Informal). *Under the same setting of Theorem 1, assume $K \leq \frac{C}{L_2(t_{max} - t_{min})}$ for some universal constant $C$. Then*

$$\underbrace{W_1(p_{t_{min}}^{\text{Restart}_\theta(K)}, p_{t_{min}})}_{\text{total error}} \leq \underbrace{B \cdot (1 - \lambda)^K TV(p_{t_{max}}^{\text{Restart}_\theta(0)}, p_{t_{max}})}_{\text{upper bound on contracted error}} + \underbrace{(K + 1) \cdot O(\delta + \epsilon_{approx})(t_{max} - t_{min})}_{\text{upper bound on additional sampling error}}$$

*where $\lambda < 1$ is the same contraction factor as Theorem 1. $O()$ hides polynomial dependency on various Lipschitz constants, dimension.*

*Proof sketch.* To bound the total error, we introduce an auxiliary process $q_{t \in [t_{\min}, t_{\max}]}^{\text{Restart}_\theta(i)}$, which initiates from true distribution $p_{t_{\max}}$ and performs the Restart iterations. This process differs from $p_{t \in [t_{\min}, t_{\max}]}^{\text{Restart}_\theta(i)}$ only in its initial distribution at $t_{\max}$ ($p_{t_{\max}}$ versus $p_{t_{\max}}^{\text{Restart}_\theta(0)}$). We bound the total error by the following triangular inequality:

$$\underbrace{W_1(p_{t_{\min}}^{\text{Restart}_\theta(K)}, p_{t_{\min}})}_{\text{total error}} \leq \underbrace{W_1(p_{t_{\min}}^{\text{Restart}_\theta(K)}, q_{t_{\min}}^{\text{Restart}_\theta(K)})}_{\text{contracted error}} + \underbrace{W_1(q_{t_{\min}}^{\text{Restart}_\theta(K)}, p_{t_{\min}})}_{\text{additional sampling error}}$$

To bound the contracted error, we construct a careful coupling process between two individual trajectories sampled from $p_{t_{\min}}^{\text{Restart}_\theta(i)}$ and $q_{t_{\min}}^{\text{Restart}_\theta(i)}$, $i = 0, \ldots, K-1$. Before these two trajectories converge, the Gaussian noise added in each Restart iteration is chosen to maximize the probability of the two trajectories mapping to an identical point, thereby maximizing the mixing rate in TV. After converging, the two processes evolve under the same Gaussian noise, and will stay converged as their drifts are the same. Lastly, we convert the TV bound to $W_1$ bound by multiplying $B$. The bound on the additional sampling error echoes the ODE analysis in Theorem 1: since the noise-injection and ODE-simulation stages are separate, we do not incur the higher discretization error of SDE. □

We defer the formal version and proof of Theorem 2 to Appendix A.1. The first term in RHS bounds the contraction on the initial error at time $t_{max}$ and the second term reflects the additional sampling error of ODE accumulated across repeated Restart iterations. Comparing the Wasserstein upper bound of SDE and ODE in Theorem 1, we make the following three observations: *(1)* Each Restart iteration has a smaller contraction factor $1 - \lambda$ compared to the one in SDE, since Restart separates the large additive noise (Eq. (3)) from the ODE (Eq. (4)). *(2)* Restart backward process (Eq. (4)) has the same order of discretization error $O(\delta)$ as the ODE, compared to $O(\sqrt{\delta})$ in SDE. Hence, the Restart allows for small NFE due to ODE-level discretization error. *(3)* The contracted error further diminishes exponentially with the number of repetitions $K$ though the additional error increases linearly with $K$. It suggests that there is a sweet spot of $K$ that strikes a balance between reducing the initial error and increasing additional sampling error. Ideally, one should pick a larger $K$ when the initial error at time $t_{max}$ greatly outweigh the incurred error in the repetitive backward process from $t_{max}$ to $t_{min}$. We provide empirical evidences in Sec 5.2.

While Theorem 1 and Theorem 2 compare the upper bounds on errors of different methods, we provide empirical validation in Section 5.1 by directly calculating these errors, showing that the Restart algorithm indeed yields a smaller total error due to its superior contraction effects. The main goal of Theorem 1 and Theorem 2 is to study how the already accumulated error changes using different samplers, and to understand their ability to self-correct the error by stochasticity. In essence, these theorems differentiate samplers based on their performance post-error accumulation. For example, by tracking the change of accumulated error, Theorem 1 shed light on the distinct "winning regions" of ODE and SDE: ODE samplers have smaller discretization error and hence excel at the small NFE regime. In contrast, SDE performs better in large NFE regime where the discretization error is negligible and its capacity to contract accumulated errors comes to the fore.

### 4.3 Practical considerations

The Restart algorithm offers several degrees of freedom, including the time interval $[t_{min}, t_{max}]$ and the number of restart iterations $K$. Here we provide a general recipe of parameter selection for practitioners, taking into account factors such as the complexity of the generative modeling tasks and the capacity of the network. Additionally, we discuss a stratified, multi-level Restart approach that further aids in reducing simulation errors along the whole trajectories for more challenging tasks.

**Where to Restart?**  Theorem 2 shows that the Restart algorithm effectively reduces the accumulated error at time $t_{max}$ by a contraction factor in the Wasserstein upper bound. These theoretical findings inspire us to position the Restart interval $[t_{min}, t_{max}]$ towards the end of the main backward process, where the accumulated error is more substantial. In addition, our empirical observations suggest that a larger time interval $t_{max} - t_{min}$ is more beneficial for weaker/smaller architectures or more challenging datasets. Even though a larger time interval increases the additional sampling error, the benefits of the contraction significantly outweighs the downside, consistent with our theoretical predictions. We leave the development of principled approaches for optimal time interval selection for future works.

**Multi-level Restart**  For challenging tasks that yield significant approximation errors, the backward trajectories may diverge substantially from the ground truth even at early stage. To prevent the ODE simulation from quickly deviating from the true trajectory, we propose implementing multiple Restart intervals in the backward process, alongside the interval placed towards the end. Empirically, we observe that a 1-level Restart is sufficient for CIFAR-10, while for more challenging datasets such as ImageNet [5], a multi-level Restart results in enhanced performance [5].

## 5 Experiments

In Sec 5.1, we first empirically verify the theoretical analysis relating to the Wasserstein upper bounds. We then evaluate the performance of different sampling algorithms on standard image generation benchmarks, including CIFAR-10 [14] and ImageNet $64 \times 64$ [5] in Sec 5.2. Lastly, we employ Restart on text-to-image generation, using Stable Diffusion model [19] pre-trained on LAION-5B [21] with resolution $512 \times 512$, in Sec 5.3.

### 5.1 Additional sampling error versus contracted error

Our proposed Restart sampling algorithm demonstrates a higher contraction effect and smaller addition sampling error compared to SDE, according to Theorem 1 and Theorem 2. Although our

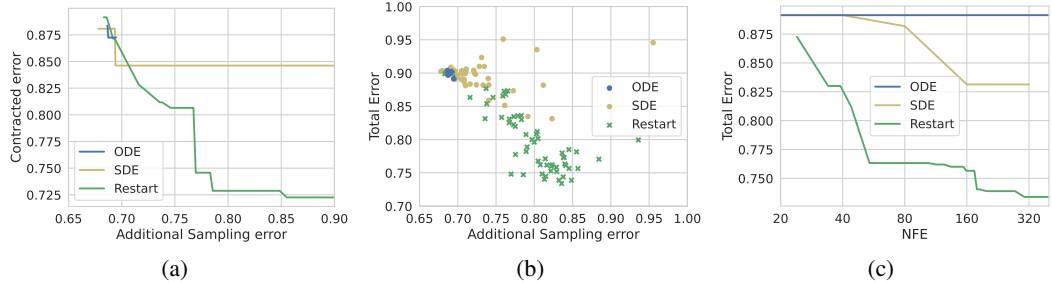

(a)  (b)  (c)

Figure 2: Additional sampling error versus **(a)** contracted error, where the Pareto frontier is plotted and **(b)** total error, where the scatter plot is provided. **(c)** Pareto frontier of NFE versus total error.

theoretical analysis compares the upper bounds of the total, contracted and additional sampling errors, we further verify their relative values through a synthetic experiment.

**Setup** We construct a 20-dimensional dataset with 2000 points sampled from a Gaussian mixture, and train a four-layer MLP to approximate the score field $\nabla_x \log p_t$. We implement the ODE, SDE, and Restart methods within a predefined time range of $[t_{\min}, t_{\max}] = [1.0, 1.5]$, where the process outside this range is conducted via the first-order ODE. To compute various error types, we define the distributions generated by three methods as outlined in the proof of Theorem 2 and directly gauge the errors at end of simulation $t = 0$ instead of $t = t_{\min}$: (1) the generated distribution as $p_0^{\text{Sampler}}$, where Sampler $\in \{\text{ODE}_\theta, \text{SDE}_\theta, \text{Restart}_\theta(K)\}$; (2) an auxiliary distribution $q_0^{\text{Sampler}}$ initiating from true distribution $p_{t_{\max}}$ at time $t_{\max}$. The only difference between $p_0^{\text{Sampler}}$ and $q_0^{\text{Sampler}}$ is their initial distribution at $t_{\max}$ ($p_{t_{\max}}^{\text{ODE}_\theta}$ versus $p_{t_{\max}}$); and (3) the true data distribution $p_0$. In line with Theorem 2, we use Wasserstein-1 distance $W_1(p_0^{\text{Sampler}}, q_0^{\text{Sampler}})$ / $W_1(q_0^{\text{Sampler}}, p_0)$ to measure the contracted error / additional sampling error, respectively. Ultimately, the total error corresponds to $W_1(p_0^{\text{Sampler}}, p_0)$. Detailed information about dataset, metric and model can be found in the Appendix C.5.

**Results** In our experiment, we adjust the parameters for all three processes and calculate the total, contracted, and additional sampling errors across all parameter settings. Figure 2(a) depicts the Pareto frontier of additional sampling error versus contracted error. We can see that Restart consistently achieves lower contracted error for a given level of additional sampling error, compared to both the ODE and SDE methods, as predicted by theory. In Figure 2(b), we observe that the Restart method obtains a smaller total error within the additional sampling error range of $[0.8, 0.85]$. During this range, Restart also displays a strictly reduced contracted error, as illustrated in Figure 2(a). This aligns with our theoretical analysis, suggesting that the Restart method offers a smaller total error due to its enhanced contraction effects. From Figure 2(c), Restart also strikes an better balance between efficiency and quality, as it achieves a lower total error at a given NFE.

## 5.2 Experiments on standard benchmarks

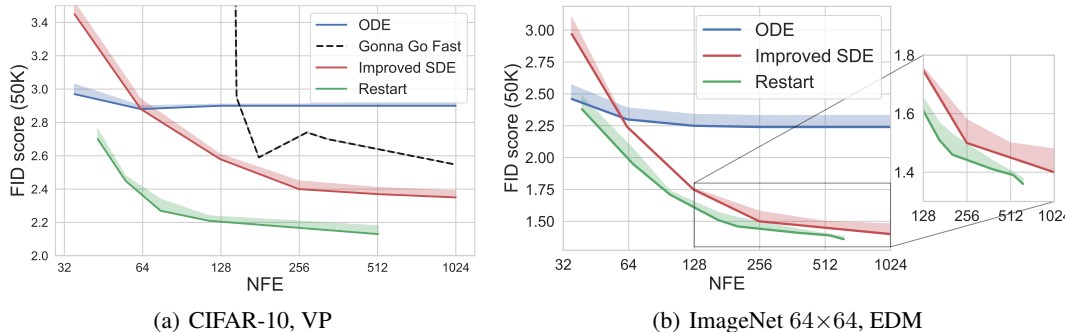

(a) CIFAR-10, VP  (b) ImageNet $64{\times}64$, EDM

Figure 3: FID versus NFE on **(a)** unconditional generation on CIFAR-10 with VP; **(b)** class-conditional generation on ImageNet with EDM.

To evaluate the sample quality and inference speed, we report the FID score [7] (lower is better) on 50K samplers and the number of function evaluations (NFE). We borrow the pretrained VP/EDM/PFGM++

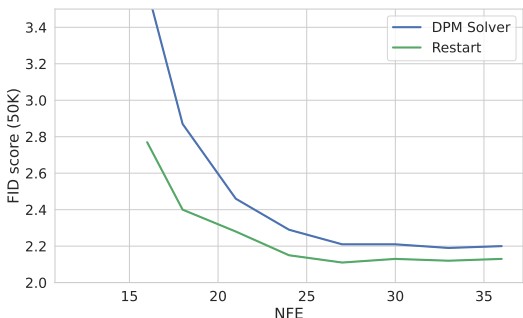

Figure 4: CIFAR-10, VP, in the low NFE regime. Restart consistently outperforms the DPM-Solver with an NFE ranging from 16 to 36.

models on CIFAR-10 or ImageNet $64 \times 64$ from [13, 28]. We also use the EDM discretization scheme [13] (see Appendix B.1 for details) during sampling.

For the proposed Restart sampler, the hyperparameters include the number of steps in the main/Restart backward processes, the number of Restart iteration $K$, as well as the time interval $[t_{\min}, t_{\max}]$. We pick the $t_{\min}$ and $t_{\max}$ from the list of time steps in EDM discretization scheme with a number of steps 18. For example, for CIFAR-10 (VP) with NFE=75, we choose $t_{\min}$=0.06, $t_{\max}$=0.30, $K$=10, where 0.30/0.06 is the 12th/14th time step in the EDM scheme. We also adopt EDM scheme for the Restart backward process in $[t_{\min}, t_{\max}]$. In addition, we apply the multi-level Restart strategy (Sec 4.3) to mitigate the error at early time steps for the more challenging ImageNet $64 \times 64$. We provide the detailed Restart configurations in Appendix C.2.

For SDE, we compare with the previously best-performing stochastic samplers proposed by [13] (**Improved SDE**). We use their optimal hyperparameters for each dataset. We also report the FID scores of the adaptive SDE [12] (**Gonna Go Fast**) on CIFAR-10 (VP). Since the vanilla reverse-diffusion SDE [23] has a significantly higher FID score, we omit its results from the main charts and defer them to Appendix D. For ODE samplers, we compare with the Heun's 2nd order method [2] (**Heun**), which arguably provides an excellent trade-off between discretization errors and NFE [13]. To ensure a fair comparison, we use Heun's method as the sampler in the main/Restart backward processes in Restart.

We report the FID score versus NFE in Figure 3(a) and Table 1 on CIFAR-10, and Figure 3(b) on ImageNet $64 \times 64$ with EDM. Our main findings are: **(1)** Restart outperforms other SDE or ODE samplers in balancing quality and speed, across datasets and models. As demonstrated in the figures, Restart achieves a 10-fold / 2-fold acceleration compared to previous best SDE results on CIFAR-10 (VP) / ImageNet $64 \times 64$ (EDM) at the same FID score. In comparison to ODE sampler (Heun), Restart obtains a better FID score, with the gap increasing significantly with NFE. **(2)** For stronger models such as EDM and PFGM++, Restart further improve over the ODE baseline on CIFAR-10. In contrast, the Improved SDE negatively impacts performance of EDM, as also observed in [13]. It suggests that Restart incorporates stochasticity more effectively. **(3)** Restart establishes new state-of-the-art FID scores for UNet architectures without additional training. In particular, Restart achieves FID scores of 1.36 on class-cond. ImageNet $64 \times 64$ with EDM, and 1.88 on uncond. CIFAR-10 with PFGM++.

**To further validate that Restart can be applied in low NFE regime**, we show that one can employ faster ODE solvers such as the **DPM-solver** [16] to further accelerate Restart. Fig. 4 shows that the Restart consistently outperforms the DPM-Solver with an NFE ranging from 16 to 36. This demonstrates Restart's capability to excel over ODE samplers, even in the small NFE regime. It also suggests that Restart can consistently improve other ODE samplers, not limited to the DDIM, Heun. Surprisingly, when paired with the DPM-Solver, Restart achieves an FID score of 2.11 on VP setting when NFE is 30, which is significantly

Table 1: Uncond. CIFAR-10 with EDM and PFGM++

|  | NFE | FID |
|---|---|---|
| *EDM-VP* [13] | | |
| ODE (Heun) | 63 | 1.97 |
|  | 35 | 1.97 |
| Improved SDE | 63 | 2.27 |
|  | 35 | 2.45 |
| Restart | 43 | **1.90** |
| *PFGM++* [28] | | |
| ODE (Heun) | 63 | 1.91 |
|  | 35 | 1.91 |
| Restart | 43 | **1.88** |

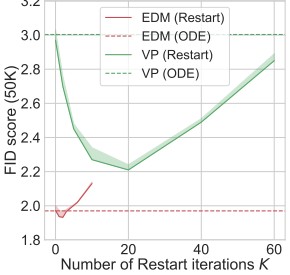

Figure 5: FID score with a varying number of Restart iterations $K$.

lower than any previous numbers (even lower than the SDE sampler with an NFE greater than 1000 in [23]), and make VP model on par with the performance with more advanced models (such as EDM).

Theorem 4 shows that each Restart iteration reduces the contracted errors while increasing the additional sampling errors in the backward process. In Fig. 5, we explore the choice of the number of Restart iterations $K$ on CIFAR-10. We find that FID score initially improves and later worsens with increasing iterations $K$, with a smaller turning point for stronger EDM model. This supports the theoretical analysis that sampling errors will eventually outweigh the contraction benefits as $K$ increases, and EDM only permits fewer Restart iterations due to smaller accumulated errors. It also suggests that, as a rule of thumb, we should apply greater Restart strength (*e.g.*, larger $K$) for weaker or smaller architectures and vice versa.

## 5.3 Experiments on large-scale text-to-image model

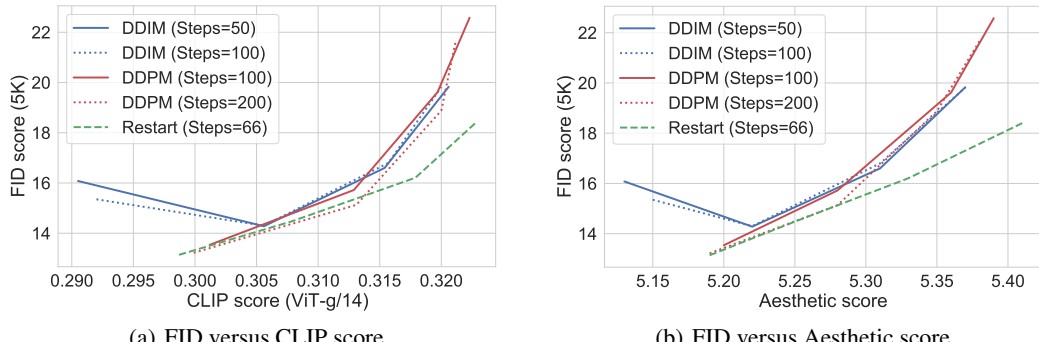

(a) FID versus CLIP score      (b) FID versus Aesthetic score

Figure 6: FID score versus **(a)** CLIP ViT-g/14 score and **(b)** Aesthetic score for text-to-image generation at $512 \times 512$ resolution, using Stable Diffusion v1.5 with a varying classifier-free guidance weight $w = 2, 3, 5, 8$.

We further apply Restart to the text-to-image Stable Diffusion v1.5 [2] pre-trained on LAION-5B [21] at a resolution of $512 \times 512$. We employ the commonly used classifier-free guidance [8, 20] for sampling, wherein each sampling step entails two function evaluations – the conditional and unconditional predictions. Following [18, 20], we use the COCO [15] validation set for evaluation. We assess text-image alignment using the CLIP score [6] with the open-sourced ViT-g/14 [11], and measure diversity via the FID score. We also evaluate visual quality through the Aesthetic score, as rated by the LAION-Aesthetics Predictor V2 [24]. Following [17], we compute all evaluation metrics using 5K captions randomly sampled from the validation set and plot the trade-off curves between CLIP/Aesthetic scores and FID score, with the classifier-free guidance weight $w$ in $\{2, 3, 5, 8\}$.

We compare with commonly used ODE sampler DDIM [22] and the stochastic sampler DDPM [9]. For Restart, we adopt the DDIM solver with 30 steps in the main backward process, and Heun in the Restart backward process, as we empirically find that Heun performs better than DDIM in the Restart. In addition, we select different sets of the hyperparameters for each guidance weight. For instance, when $w = 8$, we use $[t_{\min}, t_{\max}] = [0.1, 2]$, $K = 2$ and 10 steps in Restart backward process. We defer the detailed Restart configuration to Appendix C.2, and the results of Heun to Appendix D.1.

As illustrated in Fig. 6(a) and Fig. 6(b), Restart achieves better FID scores in most cases, given the same CLIP/Aesthetic scores, using only 132 function evaluations (*i.e.*, 66 sampling steps). Remarkably, Restart achieves substantially lower FID scores than other samplers when CLIP/Aesthetic scores are high (*i.e.*, with larger $w$ values). Conversely, Restart generally obtains a better text-image alignment/visual quality given the same FID. We also observe that DDPM generally obtains comparable performance with Restart in FID score when CLIP/Aesthetic scores are low, with Restart being more time-efficient. These findings suggest that Restart balances diversity (FID score) against text-image alignment (CLIP score) or visual quality (Aesthetic score) more effectively than previous samplers.

In Fig. 7, we visualize the images generated by Restart, DDIM and DDPM with $w = 8$. Compared to DDIM, the Restart generates images with superior details (*e.g.*, the rendition of duck legs by DDIM is less accurate) and visual quality. Compared to DDPM, Restart yields more photo-realistic images (*e.g.*, the astronaut). We provide extended of text-to-image generated samples in Appendix E.

---

[2] https://huggingface.co/runwayml/stable-diffusion-v1-5

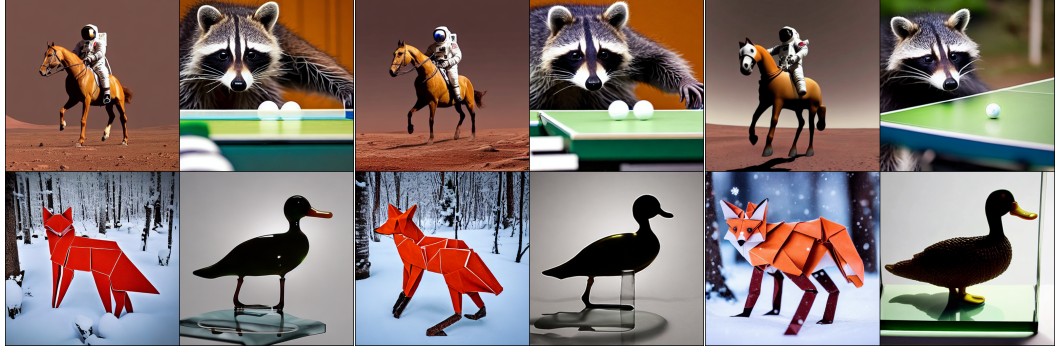

| (a) Restart (Steps=66) | (b) DDIM (Steps=100) | (c) DDPM (Steps=100) |

Figure 7: Visualization of generated images with classifier-free guidance weight $w = 8$, using four text prompts ("A photo of an astronaut riding a horse on mars.", "A raccoon playing table tennis", "Intricate origami of a fox in a snowy forest" and "A transparent sculpture of a duck made out of glass") and the **same** random seeds.

## 6 Conclusion and future direction

In this paper, we introduce the Restart sampling for generative processes involving differential equations, such as diffusion models and PFGMs. By interweaving a forward process that adds a significant amount of noise with a corresponding backward ODE, Restart harnesses and even enhances the individual advantages of both ODE and SDE. Theoretically, Restart provides greater contraction effects of stochasticity while maintaining ODE-level discretization error. Empirically, Restart achieves a superior balance between quality and time, and improves the text-image alignment/visual quality and diversity trade-off in the text-to-image Stable Diffusion models.

A current limitation of the Restart algorithm is the absence of a principled way for hyperparameters selection, including the number of iterations $K$ and the time interval $[t_{\min}, t_{\max}]$. At present, we adjust these parameters based on the heuristic that weaker/smaller models, or more challenging tasks, necessitate a stronger Restart strength. In the future direction, we anticipate developing a more principled approach to automating the selection of optimal hyperparameters for Restart based on the error analysis of models, in order to fully unleash the potential of the Restart framework.

## Acknowledgements

YX and TJ acknowledge support from MIT-DSTA Singapore collaboration, from NSF Expeditions grant (award 1918839) "Understanding the World Through Code", and from MIT-IBM Grand Challenge project. Xiang Cheng acknowledges support from NSF CCF-2112665 (TILOS AI Research Institute).

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

# Appendix

## A  Proofs of Main Theoretical Results

In this section, we provide proofs of our main results. We define below some crucial notations which we will use throughout. We use $\text{ODE}(\dots)$ to denote the backwards ODE under exact score $\nabla \log p_t(x)$. More specifically, given any $x \in \mathbb{R}^d$ and $s > r > 0$, let $x_t$ denote the solution to the following ODE:

$$dx_t = -t\nabla \log p_t(x_t)dt. \tag{5}$$

$\text{ODE}(x, s \to r)$ is defined as "the value of $x_r$ when initialized at $x_s = x$". It will also be useful to consider a "time-discretized ODE with drift $ts_\theta(x, t)$": let $\delta$ denote the discretization step size and let $k$ denote any integer. Let $\delta$ denote a step size, let $\overline{x}_t$ denote the solution to

$$d\overline{x}_t = -ts_\theta(x_{k\delta}, k\delta)dt, \tag{6}$$

where for any $t$, $k$ is the unique integer such that $t \in ((k-1)\delta, k\delta]$. We verify that the dynamics of Eq. (6) is equivalent to the following discrete-time dynamics for $t = k\delta, k \in \mathbb{Z}$:

$$\overline{x}_{(k-1)\delta} = \overline{x}_{k\delta} - \frac{1}{2}\left(((k-1)\delta)^2 - (k\delta)^2\right)s_\theta(x_{k\delta}, k\delta).$$

We similarly denote the value of $\overline{x}_r$ when initialized at $\overline{x}_s = x$ as $\text{ODE}_\theta(x, s \to r)$. Analogously, we let $\text{SDE}(x, s \to r)$ and $\text{SDE}_\theta(x, s \to r)$ denote solutions to

$$dy_t = -2t\nabla \log p_t(y_t)dt + \sqrt{2t}dB_t$$
$$d\overline{y}_t = -2ts_\theta(\overline{y}_t, t)dt + \sqrt{2t}dB_t$$

respectively. Finally, we will define the $\text{Restart}_\theta$ process as follows:

$$\begin{aligned} &(\text{Restart}_\theta \text{ forward process}) & x_{t_{\max}}^{i+1} &= x_{t_{\min}}^i + \varepsilon_{t_{\min} \to t_{\max}}^i \\ &(\text{Restart}_\theta \text{ backward process}) & x_{t_{\min}}^{i+1} &= \text{ODE}_\theta(x_{t_{\max}}^{i+1}, t_{\max} \to t_{\min}), \end{aligned} \tag{7}$$

where $\varepsilon_{t_{\min} \to t_{\max}}^i \sim \mathcal{N}\left(\mathbf{0}, \left(t_{\max}^2 - t_{\min}^2\right)I\right)$. We use $\text{Restart}_\theta(x, K)$ to denote $x_{t_{\min}}^K$ in the above processes, initialized at $x_{t_{\min}}^0 = x$. In various theorems, we will refer to a function $Q(r) : \mathbb{R}^+ \to [0, 1/2)$, defined as the Gaussian tail probability $Q(r) = Pr(a \geq r)$ for $a \sim \mathcal{N}(0, 1)$.

### A.1  Main Result

**Theorem 3.** *[Formal version of Theorem 1] Let $t_{max}$ be the initial noise level. Let the initial random variables $\overline{x}_{t_{max}} = \overline{y}_{t_{max}}$, and*

$$\begin{aligned} \overline{x}_{t_{min}} &= ODE_\theta(\overline{x}_{t_{max}}, t_{max} \to t_{min}) \\ \overline{y}_{t_{min}} &= SDE_\theta(\overline{y}_{t_{max}}, t_{max} \to t_{min}), \end{aligned}$$

*Let $p_t$ denote the true population distribution at noise level $t$. Let $p_t^{ODE_\theta}, p_t^{SDE_\theta}$ denote the distributions for $x_t, y_t$ respectively. Assume that for all $x, y, s, t$, $s_\theta(x, t)$ satisfies $\|ts_\theta(x, t) - ts_\theta(x, s)\| \leq L_0|s - t|$, $\|ts_\theta(x, t)\| \leq L_1$, $\|ts_\theta(x, t) - ts_\theta(y, t)\| \leq L_2\|x - y\|$, and the approximation error $\|ts_\theta(x, t) - t\nabla \log p_t(x)\| \leq \epsilon_{approx}$. Assume in addition that $\forall t \in [t_{min}, t_{max}]$, $\|x_t\| < B/2$ for any $x_t$ in the support of $p_t$, $p_t^{ODE_\theta}$ or $p_t^{SDE_\theta}$, and $K \leq \frac{C}{L_2(t_{max}-t_{min})}$ for some universal constant $C$. Then*

$$\begin{aligned} W_1(p_{t_{min}}^{ODE_\theta}, p_{t_{min}}) &\leq B \cdot TV\left(p_{t_{max}}^{ODE_\theta}, p_{t_{max}}\right) \\ &\quad + e^{L_2(t_{max}-t_{min})} \cdot (\delta(L_2L_1 + L_0) + \epsilon_{approx})(t_{max} - t_{min}) \end{aligned} \tag{8}$$

$$\begin{aligned} W_1(p_{t_{min}}^{SDE_\theta}, p_{t_{min}}) &\leq B \cdot \left(1 - \lambda e^{-BL_1/t_{min}-L_1^2 t_{max}^2/t_{min}^2}\right) TV(p_{t_{max}}^{SDE_\theta}, p_{t_{max}}) \\ &\quad + e^{2L_2(t_{max}-t_{min})}\left(\epsilon_{approx} + \delta L_0 + L_2\left(\delta L_1 + \sqrt{2\delta dt_{max}}\right)\right)(t_{max} - t_{min}) \end{aligned} \tag{9}$$

*where $\lambda := 2Q\left(\frac{B}{2\sqrt{t_{max}^2 - t_{min}^2}}\right)$.*

*Proof.* Let us define $x_{t_{\max}} \sim p_{t_{\max}}$, and let $x_{t_{\min}} = \text{ODE}(x_{t_{\max}}, t_{\max} \to t_{\min})$. We verify that $x_{t_{\min}}$ has density $p_{t_{\min}}$. Let us also define $\hat{x}_{t_{\min}} = \text{ODE}_\theta(x_{t_{\max}}, t_{\max} \to t_{\min})$. We would like to bound the Wasserstein distance between $\bar{x}_{t_{\min}}$ and $x_{t_{\min}}$ (*i.e.*, $p_{t_{\min}}^{\text{ODE}_\theta}$ and $p_{t_{\min}}$), by the following triangular inequality:

$$W_1(\bar{x}_{t_{\min}}, x_{t_{\min}}) \leq W_1(\bar{x}_{t_{\min}}, \hat{x}_{t_{\min}}) + W_1(\hat{x}_{t_{\min}}, x_{t_{\min}}) \tag{10}$$

By Lemma 2, we know that

$$\|\hat{x}_{t_{\min}} - x_{t_{\min}}\| \leq e^{(t_{\max} - t_{\min})L_2}\left(\delta(L_2 L_1 + L_0) + \epsilon_{approx}\right)(t_{\max} - t_{\min}),$$

where we use the fact that $\|\hat{x}_{t_{\max}} - x_{t_{\max}}\| = 0$. Thus we immediately have

$$W_1(\hat{x}_{t_{\min}}, x_{t_{\min}}) \leq e^{(t_{\max} - t_{\min})L_2}\left(\delta(L_2 L_1 + L_0) + \epsilon_{approx}\right)(t_{\max} - t_{\min}) \tag{11}$$

On the other hand,

$$\begin{aligned} W_1(\hat{x}_{t_{\min}}, \bar{x}_{t_{\min}}) &\leq B \cdot TV\left(\hat{x}_{t_{\min}}, \bar{x}_{t_{\min}}\right) \\ &\leq B \cdot TV\left(\hat{x}_{t_{\max}}, \bar{x}_{t_{\max}}\right) \end{aligned} \tag{12}$$

where the last equality is due to the data-processing inequality. Combining Eq. (11), Eq. (12) and the triangular inequality Eq. (10), we arrive at the upper bound for ODE (Eq. (8)). The upper bound for SDE (Eq. (9)) shares a similar proof approach. First, let $y_{t_{\max}} \sim p_{t_{\max}}$. Let $\hat{y}_{t_{\min}} = \text{SDE}_\theta(y_{t_{\max}}, t_{\max} \to t_{\min})$. By Lemma 5,

$$TV\left(\hat{y}_{t_{\min}}, \bar{y}_{t_{\min}}\right) \leq \left(1 - 2Q\left(\frac{B}{2\sqrt{t_{\max}^2 - t_{\min}^2}}\right) \cdot e^{-BL_1/t_{\min} - L_1^2 t_{\max}^2 / t_{\min}^2}\right) \cdot TV\left(\hat{y}_{t_{\max}}, \bar{y}_{t_{\max}}\right)$$

On the other hand, by Lemma 4,

$$\mathbb{E}\left[\|\hat{y}_{t_{\min}} - y_{t_{\min}}\|\right] \leq e^{2L_2(t_{\max} - t_{\min})}\left(\epsilon_{approx} + \delta L_0 + L_2\left(\delta L_1 + \sqrt{2\delta d t_{\max}}\right)\right)(t_{\max} - t_{\min}).$$

The SDE triangular upper bound on $W_1(\bar{y}_{t_{\min}}, y_{t_{\min}})$ follows by multiplying the first inequality by $B$ (to bound $W_1(\bar{y}_{t_{\min}}, \hat{y}_{t_{\min}})$) and then adding the second inequality (to bound $W_1(y_{t_{\min}}, \hat{y}_{t_{\min}})$). Notice that by definition, $TV\left(\hat{y}_{t_{\max}}, \bar{y}_{t_{\max}}\right) = TV\left(y_{t_{\max}}, \bar{y}_{t_{\max}}\right)$. Finally, because of the assumption that $K \leq \frac{C}{L_2(t_{\max} - t_{\min})}$ for some universal constant, we summarize the second term in the Eq. (8) and Eq. (9) into the big $O$ in the informal version Theorem 1. □

**Theorem 4.** *[Formal version of Theorem 2] Consider the same setting as Theorem 3. Let $p_{t_{min}}^{Restart_\theta, i}$ denote the distributions after $i^{th}$ Restart iteration, i.e., the distribution of $\bar{x}_{t_{min}}^i = Restart_\theta(\bar{x}_{t_{min}}^0, i)$. Given initial $\bar{x}_{t_{max}}^0 \sim p_{t_{max}}^{Restart,0}$, let $\bar{x}_{t_{min}}^0 = ODE_\theta(\bar{x}_{t_{max}}^0, t_{max} \to t_{min})$. Then*

$$\begin{aligned} W_1(p_{t_{min}}^{Restart_\theta, K}, p_{t_{min}}) \leq &\underbrace{B \cdot (1 - \lambda)^K TV(p_{t_{max}}^{Restart,0}, p_{t_{max}})}_{\text{upper bound on contracted error}} \\ &+ \underbrace{e^{(K+1)L_2(t_{max} - t_{min})}(K + 1)\left(\delta(L_2 L_1 + L_0) + \epsilon_{approx}\right)(t_{max} - t_{min})}_{\text{upper bound on additional sampling error}} \end{aligned} \tag{13}$$

*where* $\lambda = 2Q\left(\frac{B}{2\sqrt{t_{max}^2 - t_{min}^2}}\right)$.

*Proof.* Let $x_{t_{\max}}^0 \sim p_{t_{\max}}$. Let $x_{t_{\min}}^K = \text{Restart}(x_{t_{\min}}^0, K)$. We verify that $x_{t_{\min}}^K$ has density $p_{t_{\min}}$. Let us also define $\hat{x}_{t_{\min}}^0 = \text{ODE}_\theta(x_{t_{\max}}^0, t_{\max} \to t_{\min})$ and $\hat{x}_{t_{\min}}^K = \text{Restart}_\theta(\hat{x}_{t_{\min}}^0, K)$.

By Lemma 1,

$$\begin{aligned} TV\left(\bar{x}_{t_{\min}}^K, \hat{x}_{t_{\min}}^K\right) &\leq \left(1 - 2Q\left(\frac{B}{2\sqrt{t_{\max}^2 - t_{\min}^2}}\right)\right)^K TV\left(\bar{x}_{t_{\min}}^0, \hat{x}_{t_{\min}}^0\right) \\ &\leq \left(1 - 2Q\left(\frac{B}{2\sqrt{t_{\max}^2 - t_{\min}^2}}\right)\right)^K TV\left(\bar{x}_{t_{\max}}^0, \hat{x}_{t_{\max}}^0\right) \\ &= \left(1 - 2Q\left(\frac{B}{2\sqrt{t_{\max}^2 - t_{\min}^2}}\right)\right)^K TV\left(\bar{x}_{t_{\max}}^0, x_{t_{\max}}^0\right) \end{aligned}$$

The second inequality holds by data processing inequality. The above can be used to bound the 1-Wasserstein distance as follows:

$$W_1\left(\overline{x}^K_{t_{\min}}, \hat{x}^K_{t_{\min}}\right) \le B \cdot TV\left(\overline{x}^K_{t_{\min}}, \hat{x}^K_{t_{\min}}\right) \le \left(1 - 2Q\left(\frac{B}{2\sqrt{t^2_{\max} - t^2_{\min}}}\right)\right)^K TV\left(\overline{x}^0_{t_{\max}}, x^0_{t_{\max}}\right)$$

(14)

On the other hand, using Lemma 3,

$$W_1\left(x^K_{t_{\min}}, \hat{x}^K_{t_{\min}}\right) \le \left\|x^K_{t_{\min}} - \hat{x}^K_{t_{\min}}\right\|$$
$$\le e^{(K+1)L_2(t_{\max} - t_{\min})}(K+1)\left(\delta(L_2 L_1 + L_0) + \epsilon_{approx}\right)(t_{\max} - t_{\min}) \quad (15)$$

We arrive at the result by combining the two bounds above (Eq. (14), Eq. (15)) with the following triangular inequality,

$$W_1(\overline{x}^K_{t_{\min}}, x^K_{t_{\min}}) \le W_1(\overline{x}^K_{t_{\min}}, \hat{x}^K_{t_{\min}}) + W_1(\hat{x}^K_{t_{\min}}, x^K_{t_{\min}})$$

$\square$

## A.2 Mixing under Restart with exact ODE

**Lemma 1.** *Consider the same setup as Theorem 4. Consider the Restart$_\theta$ process defined in equation 7. Let*

$$x^i_{t_{min}} = Restart_\theta(x^0_{t_{min}}, i)$$
$$y^i_{t_{min}} = Restart_\theta(y^0_{t_{min}}, i).$$

*Let $p_t^{Restart_\theta(i)}$ and $q_t^{Restart_\theta(i)}$ denote the densities of $x^i_t$ and $y^i_t$ respectively. Then*

$$TV\left(p_{t_{min}}^{Restart_\theta(K)}, q_{t_{min}}^{Restart_\theta(K)}\right) \le (1 - \lambda)^K TV\left(p_{t_{min}}^{Restart_\theta(0)}, q_{t_{min}}^{Restart_\theta(0)}\right),$$

*where $\lambda = 2Q\left(\frac{B}{2\sqrt{t^2_{max} - t^2_{min}}}\right)$.*

*Proof.* Conditioned on $x^i_{t_{\min}}, y^i_{t_{\min}}$, let $x^{i+1}_{t_{\max}} = x^i_{t_{\min}} + \sqrt{t^2_{\max} - t^2_{\min}}\xi^x_i$ and $y^{i+1}_{t_{\max}} = y^i_{t_{\min}} + \sqrt{t^2_{\max} - t^2_{\min}}\xi^y_i$. We now define a coupling between $x^{i+1}_{t_{\min}}$ and $y^{i+1}_{t_{\min}}$ by specifying the joint distribution over $\xi^x_i$ and $\xi^y_i$.

If $x^i_{t_{\min}} = y^i_{t_{\min}}$, let $\xi^x_i = \xi^y_i$, so that $x^{i+1}_{t_{\min}} = y^{i+1}_{t_{\min}}$. On the other hand, if $x^i_{t_{\min}} \ne y^i_{t_{\min}}$, let $x^{i+1}_{t_{\max}}$ and $y^{i+1}_{t_{\max}}$ be coupled as described in the proof of Lemma 7, with $x' = x^{i+1}_{t_{\max}}, y' = y^{i+1}_{t_{\max}}, \sigma = \sqrt{t^2_{\max} - t^2_{\min}}$. Under this coupling, we verify that,

$$\mathbb{E}\left[\mathbb{1}\left\{x^{i+1}_{t_{\min}} \ne y^{i+1}_{t_{\min}}\right\}\right]$$
$$\le \mathbb{E}\left[\mathbb{1}\left\{x^{i+1}_{t_{\max}} \ne y^{i+1}_{t_{\max}}\right\}\right]$$
$$\le \mathbb{E}\left[\left(1 - 2Q\left(\frac{\|x^i_{t_{\min}} - y^i_{t_{\min}}\|}{2\sqrt{t^2_{\max} - t^2_{\min}}}\right)\right)\mathbb{1}\left\{x^i_{t_{\min}} \ne y^i_{t_{\min}}\right\}\right]$$
$$\le \left(1 - 2Q\left(\frac{B}{2\sqrt{t^2_{\max} - t^2_{\min}}}\right)\right)\mathbb{E}\left[\mathbb{1}\left\{x^i_{t_{\min}} \ne y^i_{t_{\min}}\right\}\right].$$

Applying the above recursively,

$$\mathbb{E}\left[\mathbb{1}\left\{x^K_{t_{\min}} \ne y^K_{t_{\min}}\right\}\right] \le \left(1 - 2Q\left(\frac{B}{2\sqrt{t^2_{\max} - t^2_{\min}}}\right)\right)^K \mathbb{E}\left[\mathbb{1}\left\{x^0_{t_{\min}} \ne y^0_{t_{\min}}\right\}\right].$$

The conclusion follows by noticing that $TV\left(p_{t_{\min}}^{Restart_\theta(K)}, q_{t_{\min}}^{Restart_\theta(K)}\right) \le Pr\left(x^K_{t_{\min}} \ne y^K_{t_{\min}}\right) = \mathbb{E}\left[\mathbb{1}\left\{x^K_{t_{\min}} \ne y^K_{t_{\min}}\right\}\right]$, and by selecting the initial coupling so that $Pr\left(x^0_{t_{\min}} \ne y^0_{t_{\min}}\right) = TV\left(p_{t_{\min}}^{Restart_\theta(0)}, q_{t_{\min}}^{Restart_\theta(0)}\right)$. $\square$

### A.3 $W_1$ discretization bound

**Lemma 2** (Discretization bound for ODE). *Let $x_{t_{min}} = ODE(x_{t_{max}}, t_{max} \to t_{min})$ and let $\overline{x}_{t_{min}} = ODE_\theta(\overline{x}_{t_{max}}, t_{max} \to t_{min})$. Assume that for all $x, y, s, t$, $s_\theta(x, t)$ satisfies $\|ts_\theta(x, t) - ts_\theta(x, s)\| \leq L_0|s - t|$, $\|ts_\theta(x, t)\| \leq L_1$ and $\|ts_\theta(x, t) - ts_\theta(y, t)\| \leq L_2\|x - y\|$. Then*

$$\|x_{t_{min}} - \overline{x}_{t_{min}}\| \leq e^{(t_{max} - t_{min})L_2}\left(\|x_{t_{max}} - \overline{x}_{t_{max}}\| + (\delta(L_2L_1 + L_0) + \epsilon_{approx})(t_{max} - t_{min})\right)$$

*Proof.* Consider some fixed arbitrary $k$, and recall that $\delta$ is the step size. Recall that by definition of ODE and $ODE_\theta$, for $t \in ((k-1)\delta, k\delta]$,

$$dx_t = -t\nabla \log p_t(x_t)dt$$
$$d\overline{x}_t = -ts_\theta(\overline{x}_{k\delta}, k\delta)dt.$$

For $t \in [t_{\min}, t_{\max}]$, let us define a time-reversed process $x_t^\leftarrow := x_{-t}$. Let $v(x, t) := \nabla \log p_{-t}(x)$. Then for $t \in [-t_{\max}, -t_{\min}]$

$$dx_t^\leftarrow = tv(x_t^\leftarrow, t)ds.$$

Similarly, define $\overline{x}_t^\leftarrow := \overline{x}_{-t}$ and $\overline{v}(x, t) := s_\theta(x, -t)$. It follows that

$$d\overline{x}_t^\leftarrow = t\overline{v}(\overline{x}_{k\delta}^\leftarrow, k\delta)ds,$$

where $k$ is the unique (negative) integer satisfying $t \in [k\delta, (k+1)\delta)$. Following these definitions,

$$\frac{d}{dt}\|x_t^\leftarrow - \overline{x}_t^\leftarrow\|$$
$$\leq \|tv(x_t^\leftarrow, t) - t\overline{v}(x_t^\leftarrow, t)\|$$
$$\quad + \|t\overline{v}(x_t^\leftarrow, t) - t\overline{v}(\overline{x}_t^\leftarrow, t)\|$$
$$\quad + \|t\overline{v}(\overline{x}_t^\leftarrow, t) - t\overline{v}(\overline{x}_t^\leftarrow, k\delta)\|$$
$$\quad + \|t\overline{v}(\overline{x}_t^\leftarrow, k\delta) - t\overline{v}(\overline{x}_{k\delta}^\leftarrow, k\delta)\|$$
$$\leq \epsilon_{approx} + L_2\|x_t^\leftarrow - \overline{x}_t^\leftarrow\| + \delta L_0 + L_2\|\overline{x}_t^\leftarrow - \overline{x}_{k\delta}^\leftarrow\|$$
$$\leq \epsilon_{approx} + L_2\|x_t^\leftarrow - \overline{x}_t^\leftarrow\| + \delta L_0 + \delta L_2 L_1.$$

Applying Gronwall's Lemma over the interval $t \in [-t_{\max}, -t_{\min}]$,

$$\|x_{t_{\min}} - \overline{x}_{t_{\min}}\|$$
$$= \|x_{-t_{\min}}^\leftarrow - \overline{x}_{-t_{\min}}^\leftarrow\|$$
$$\leq e^{L_2(t_{max} - t_{min})}\left(\|x_{-t_{\max}}^\leftarrow - \overline{x}_{-t_{\max}}^\leftarrow\| + (\epsilon_{approx} + \delta L_0 + \delta L_2 L_1)(t_{max} - t_{min})\right)$$
$$= e^{L_2(t_{max} - t_{min})}\left(\|x_{t_{\max}} - \overline{x}_{t_{\max}}\| + (\epsilon_{approx} + \delta L_0 + \delta L_2 L_1)(t_{max} - t_{min})\right).$$

$\square$

**Lemma 3.** *Given initial $x_{t_{max}}^0$, let $x_{t_{min}}^0 = ODE(x_{t_{max}}^0, t_{max} \to t_{min})$, and let $\hat{x}_{t_{min}}^0 = ODE_\theta(x_{t_{max}}^0, t_{max} \to t_{min})$. We further denote the variables after $K$ Restart iterations as $x_{t_{min}}^K = Restart(x_{t_{min}}^0, K)$ and $\hat{x}_{t_{min}}^K = Restart_\theta(\hat{x}_{t_{min}}^0, K)$, with true field and learned field respectively. Then there exists a coupling between $x_{t_{min}}^K$ and $\hat{x}_{t_{min}}^K$ such that*

$$\|x_{t_{min}}^K - \hat{x}_{t_{min}}^K\| \leq e^{(K+1)L_2(t_{max} - t_{min})}(K+1)(\delta(L_2L_1 + L_0) + \epsilon_{approx})(t_{max} - t_{min}).$$

*Proof.* We will couple $x_{t_{\min}}^i$ and $\hat{x}_{t_{\min}}^i$ by using the same noise $\varepsilon_{t_{\min} \to t_{\max}}^i$ in the Restart forward process for $i = 0 \ldots K - 1$ (see Eq. (7)). For any $i$, let us also define $y_{t_{\min}}^{i,j} := Restart_\theta(x_{t_{\min}}^i, j - i)$, and this process uses the same noise $\varepsilon_{t_{\min} \to t_{\max}}^i$ as previous ones. From this definition, $y_{t_{\min}}^{K,K} = x_{t_{\min}}^K$. We can thus bound

$$\|x_{t_{\min}}^K, \hat{x}_{t_{\min}}^K\| \leq \|y_{t_{\min}}^{0,K} - \hat{x}_{t_{\min}}^K\| + \sum_{i=0}^{K-1}\|y_{t_{\min}}^{i,K} - y_{t_{\min}}^{i+1,K}\| \tag{16}$$

Using the assumption that $ts_\theta(\cdot, t)$ is $L_2$ Lipschitz,

$$\left\| y^{0,i+1}_{t_{\min}} - \hat{x}^{i+1}_{t_{\min}} \right\|$$
$$= \left\| \text{ODE}_\theta(y^{0,i}_{t_{\max}}, t_{\max} \to t_{\min}) - \text{ODE}_\theta(\hat{x}^i_{t_{\max}}, t_{\max} \to t_{\min}) \right\|$$
$$\leq e^{L_2(t_{\max}-t_{\min})} \left\| y^{0,i}_{t_{\max}} - \hat{x}^i_{t_{\max}} \right\|$$
$$= e^{L_2(t_{\max}-t_{\min})} \left\| y^{0,i}_{t_{\min}} - \hat{x}^i_{t_{\min}} \right\|,$$

where the last equality is because we add the same additive Gaussian noise $\varepsilon^i_{t_{\min}\to t_{\max}}$ to $y^{0,i}_{t_{\min}}$ and $\hat{x}^i_{t_{\min}}$ in the Restart forward process. Applying the above recursively, we get

$$\left\| y^{0,K}_{t_{\min}} - \hat{x}^K_{t_{\min}} \right\| \leq e^{KL_2(t_{\max}-t_{\min})} \left\| y^{0,0}_{t_{\min}} - \hat{x}^0_{t_{\min}} \right\|$$
$$\leq e^{KL_2(t_{\max}-t_{\min})} \left\| x^0_{t_{\min}} - \hat{x}^0_{t_{\min}} \right\|$$
$$\leq e^{(K+1)L_2(t_{\max}-t_{\min})} \left( \delta(L_2 L_1 + L_0) + \epsilon_{approx} \right) (t_{\max} - t_{\min}), \qquad (17)$$

where the last line follows by Lemma 2 when setting $x_{t_{\max}} = \bar{x}_{t_{\max}}$. We will now bound $\left\| y^{i,K}_{t_{\min}} - y^{i+1,K}_{t_{\min}} \right\|$ for some $i \leq K$. It follows from definition that

$$y^{i,i+1}_{t_{\min}} = \text{ODE}_\theta \left( x^i_{t_{\max}}, t_{\max} \to t_{\min} \right)$$
$$y^{i+1,i+1}_{t_{\min}} = x^{i+1}_{t_{\min}} = \text{ODE} \left( x^i_{t_{\max}}, t_{\max} \to t_{\min} \right).$$

By Lemma 2,

$$\left\| y^{i,i+1}_{t_{\min}} - y^{i+1,i+1}_{t_{\min}} \right\| \leq e^{L_2(t_{\max}-t_{\min})} \left( \delta(L_2 L_1 + L_0) + \epsilon_{approx} \right) (t_{\max} - t_{\min})$$

For the remaining steps from $i + 2 \ldots K$, both $y^{i,\cdot}$ and $y^{i+1,\cdot}$ evolve with $\text{ODE}_\theta$ in each step. Again using the assumption that $ts_\theta(\cdot, t)$ is $L_2$ Lipschitz,

$$\left\| y^{i,K}_{t_{\min}} - y^{i+1,K}_{t_{\min}} \right\| \leq e^{(K-i)L_2(t_{\max}-t_{\min})} \left( \delta(L_2 L_1 + L_0) + \epsilon_{approx} \right) (t_{\max} - t_{\min})$$

Summing the above for $i = 0 \ldots K - 1$, and combining with Eq. (16) and Eq. (17) gives

$$\left\| x^K_{t_{\min}} - \hat{x}^K_{t_{\min}} \right\| \leq e^{(K+1)L_2(t_{\max}-t_{\min})}(K + 1) \left( \delta(L_2 L_1 + L_0) + \epsilon_{approx} \right) (t_{\max} - t_{\min}).$$

$\square$

**Lemma 4.** *Consider the same setup as Theorem 3. Let $x_{t_{min}} = SDE\left(x_{t_{max}}, t_{max} \to t_{min}\right)$ and let $\bar{x}_{t_{min}} = SDE\left(\bar{x}_{t_{max}}, t_{max} \to t_{min}\right)$. Then there exists a coupling between $x_t$ and $\bar{x}_t$ such that*

$$\mathbb{E}\left[ \|x_{t_{min}} - \bar{x}_{t_{min}}\| \right] \leq e^{2L_2(t_{max}-t_{min})} \mathbb{E}\left[ \|x_{t_{max}} - \bar{x}_{t_{max}}\| \right]$$
$$+ e^{2L_2(t_{max}-t_{min})} \left( \epsilon_{approx} + \delta L_0 + L_2 \left( \delta L_1 + \sqrt{2\delta dt_{max}} \right) \right) (t_{max} - t_{min})$$

*Proof.* Consider some fixed arbitrary $k$, and recall that $\delta$ is the stepsize. By definition of SDE and $\text{SDE}_\theta$, for $t \in ((k-1)\delta, k\delta]$,

$$dx_t = -2t\nabla \log p_t(x_t)dt + \sqrt{2t}dB_t$$
$$d\bar{x}_t = -2ts_\theta(\bar{x}_{k\delta}, k\delta)dt + \sqrt{2t}dB_t.$$

Let us define a coupling between $x_t$ and $\bar{x}_t$ by identifying their respective Brownian motions. It will be convenient to define the time-reversed processes $x^\leftarrow_t := x_{-t}$, and $\bar{x}^\leftarrow_t := \bar{x}_{-t}$, along with $v(x, t) := \nabla \log p_{-t}(x)$ and $\bar{v}(x, t) := s_\theta(x, -t)$. Then there exists a Brownian motion $B^\leftarrow_t$, such that for $t \in [-t_{\max}, -t_{\min}]$,

$$dx^\leftarrow_t = -2tv(x^\leftarrow_t, t)dt + \sqrt{-2t}dB^\leftarrow_t$$
$$d\bar{x}^\leftarrow_t = -2t\bar{v}(\bar{x}^\leftarrow_{k\delta}, k\delta)dt + \sqrt{-2t}dB^\leftarrow_t$$
$$\Rightarrow \quad d(x^\leftarrow_t - \bar{x}^\leftarrow_t) = -2t \left( v(x^\leftarrow_t, t) - \bar{v}(\bar{x}^\leftarrow_{k\delta}, k\delta) \right) dt,$$

where $k$ is the unique negative integer such that $t \in [k\delta, (k+1)\delta)$. Thus

$$
\begin{aligned}
&\frac{d}{dt}\mathbb{E}\left[\|x_t^{\leftarrow} - \overline{x}_t^{\leftarrow}\|\right] \\
&\leq 2\left(\mathbb{E}\left[\|tv(x_t^{\leftarrow}, t) - t\overline{v}(x_t^{\leftarrow}, t)\|\right] + \mathbb{E}\left[\|t\overline{v}(x_t^{\leftarrow}, t) - t\overline{v}(\overline{x}_t^{\leftarrow}, t)\|\right]\right) \\
&\qquad + 2\left(\mathbb{E}\left[\|t\overline{v}(\overline{x}_t^{\leftarrow}, t) - t\overline{v}(\overline{x}_t^{\leftarrow}, k\delta)\|\right] + \mathbb{E}\left[\|t\overline{v}(\overline{x}_t^{\leftarrow}, k\delta) - t\overline{v}(\overline{x}_{k\delta}^{\leftarrow}, k\delta)\|\right]\right) \\
&\leq 2\left(\epsilon_{approx} + L_2\mathbb{E}\left[\|x_t^{\leftarrow} - \overline{x}_t^{\leftarrow}\|\right] + \delta L_0 + L_2\mathbb{E}\left[\|\overline{x}_t^{\leftarrow} - \overline{x}_{k\delta}^{\leftarrow}\|\right]\right) \\
&\leq 2\left(\epsilon_{approx} + L_2\mathbb{E}\left[\|x_t^{\leftarrow} - \overline{x}_t^{\leftarrow}\|\right] + \delta L_0 + L_2\left(\delta L_1 + \sqrt{2\delta dt_{\max}}\right)\right).
\end{aligned}
$$

By Gronwall's Lemma,

$$
\begin{aligned}
&\mathbb{E}\left[\|x_{t_{\min}} - \overline{x}_{t_{\min}}\|\right] \\
&= \mathbb{E}\left[\|x_{-t_{\min}}^{\leftarrow} - \overline{x}_{-t_{\min}}^{\leftarrow}\|\right] \\
&\leq e^{2L_2(t_{\max} - t_{\min})}\left(\mathbb{E}\left[\|x_{-t_{\max}}^{\leftarrow} - \overline{x}_{-t_{\max}}^{\leftarrow}\|\right] + \left(\epsilon_{approx} + \delta L_0 + L_2\left(\delta L_1 + \sqrt{2\delta dt_{\max}}\right)\right)(t_{\max} - t_{\min})\right) \\
&= e^{2L_2(t_{\max} - t_{\min})}\left(\mathbb{E}\left[\|x_{t_{\max}} - \overline{x}_{t_{\max}}\|\right] + \left(\epsilon_{approx} + \delta L_0 + L_2\left(\delta L_1 + \sqrt{2\delta dt_{\max}}\right)\right)(t_{\max} - t_{\min})\right)
\end{aligned}
$$

$$\square$$

### A.4 Mixing Bounds

**Lemma 5.** *Consider the same setup as Theorem 3. Assume that $\delta \leq t_{min}$. Let*

$$
\begin{aligned}
x_{t_{min}} &= SDE_\theta\left(x_{t_{max}}, t_{max} \to t_{min}\right) \\
y_{t_{min}} &= SDE_\theta\left(y_{t_{max}}, t_{max} \to t_{min}\right).
\end{aligned}
$$

*Then there exists a coupling between $x_s$ and $y_s$ such that*

$$
TV\left(x_{t_{min}}, y_{t_{min}}\right) \leq \left(1 - 2Q\left(\frac{B}{2\sqrt{t_{max}^2 - t_{min}^2}}\right) \cdot e^{-BL_1/t_{min} - L_1^2 t_{max}^2/t_{min}^2}\right) TV\left(x_{t_{max}}, y_{t_{max}}\right)
$$

*Proof.* We will construct a coupling between $x_t$ and $y_t$. First, let $(x_{t_{\max}}, y_{t_{\max}})$ be sampled from the optimal TV coupling, *i.e.*, $Pr(x_{t_{\max}} \neq y_{t_{\max}}) = TV(x_{t_{\max}}, y_{t_{\max}})$. Recall that by definition of $SDE_\theta$, for $t \in ((k-1)\delta, k\delta]$,

$$
dx_t = -2ts_\theta(x_{k\delta}, k\delta)dt + \sqrt{2t}dB_t.
$$

Let us define a time-rescaled version of $x_t$: $\overline{x}_t := x_{t^2}$. We verify that

$$
d\overline{x}_t = -s_\theta(\overline{x}_{(k\delta)^2}, k\delta)dt + dB_t,
$$

where $k$ is the unique integer satisfying $t \in [((k-1)\delta)^2, k^2\delta^2)$. Next, we define the time-reversed process $\overline{x}_t^{\leftarrow} := \overline{x}_{-t}$, and let $v(x, t) := s_\theta(x, -t)$. We verify that there exists a Brownian motion $B_t^x$ such that, for $t \in [-t_{\max}^2, -t_{\min}^2]$,

$$
d\overline{x}_t^{\leftarrow} = v_t^x dt + dB_t^x,
$$

where $v_t^x = s_\theta(\overline{x}_{-(k\delta)^2}^{\leftarrow}, -k\delta)$, where $k$ is the unique positive integer satisfying $-t \in (((k-1)\delta)^2, (k\delta)^2]$. Let $d\overline{y}_t^{\leftarrow} = v_t^y dt + dB_t^y$, be defined analogously. For any positive integer $k$ and for any $t \in [-(k\delta)^2, -((k-1)\delta)^2)$, let us define

$$
z_t = \overline{x}_{-k^2\delta^2}^{\leftarrow} - \overline{y}_{-k^2\delta^2}^{\leftarrow} + (2k-1)\delta^2\left(v_{-(k\delta)^2}^x - v_{-(k\delta)^2}^y\right) + \left(B_t^x - B_{-(k\delta)^2}^x\right) - \left(B_t^y - B_{-(k\delta)^2}^y\right).
$$

Let $\gamma_t := \frac{z_t}{\|z_t\|}$. We will now define a coupling between $dB_t^x$ and $dB_t^y$ as

$$
dB_t^y = \left(I - 2\mathbb{1}\left\{t \leq \tau\right\}\gamma_t\gamma_t^T\right) dB_t^x,
$$

where $\mathbb{1}\{\}$ denotes the indicator function, i.e. $\mathbb{1}\{t \leq \tau\} = 1$ if $t \leq \tau$, and $\tau$ is a stopping time given by the first hitting time of $z_t = 0$. Let $r_t := \|z_t\|$. Consider some $t \in \left(-i^2\delta^2, -(i-1)^2\delta^2\right)$, and Let $j := \frac{t_{\max}}{\delta}$ (assume w.l.o.g that this is an integer), then

$$
\begin{aligned}
r_t - r_{-t_{\max}^2} &\leq \sum_{k=i}^{j} (2k-1)\delta^2 \left\| \left(v^x_{-(k\delta)^2} - v^y_{-(k\delta)^2}\right) \right\| + \int_{-t_{\max}^2}^{t} \mathbb{1}\{t \leq \tau\} 2dB_s^1 \\
&\leq \sum_{k=i}^{j} \left(k^2 - (k-1)^2\right)\delta^2 2L_1/(t_{\min}) + \int_{-t_{\max}^2}^{t} \mathbb{1}\{t \leq \tau\} 2dB_t^1 \\
&= \int_{-t_{\max}^2}^{-(i-1)\delta^2} \frac{2L_1}{t_{\min}} ds + \int_{-t_{\max}^2}^{t} \mathbb{1}\{t \leq \tau\} 2dB_s^1,
\end{aligned}
$$

where $dB_s^1 = \langle \gamma_t, dB_s^x - dB_s^y \rangle$ is a 1-dimensional Brownian motion. We also verify that

$$
\begin{aligned}
r_{-t_{\max}^2} &= \left\| z_{-t_{\max}^2} \right\| \\
&= \left\| \overline{x}^{\leftarrow}_{-t_{\max}^2} - \overline{y}^{\leftarrow}_{-t_{\max}^2} + (2j-1)\delta^2 \left(v^x_{-t_{\max}^2} - v^y_{-t_{\max}^2}\right) + \left(B_t^x - B_{-t_{\max}^2}^x\right) - \left(B_t^y - B_{-t_{\max}^2}^y\right) \right\| \\
&\leq \left\| \overline{x}^{\leftarrow}_{-t_{\max}^2} + (2j-1)\delta^2 v^x_{-t_{\max}^2} + \left(B_{-(j-1)^2\delta^2}^x - B_{-t_{\max}^2}^x\right) \right\| \\
&\quad + \left\| \overline{y}^{\leftarrow}_{-t_{\max}^2} + (2j-1)\delta^2 v^y_{-t_{\max}^2} + \left(B_{-(j-1)^2\delta^2}^x - B_t^x + B_t^y - B_{-t_{\max}^2}^y\right) \right\| \leq B
\end{aligned}
$$

where the third relation is by adding and subtracting $B_{-(j-1)^2\delta^2}^x - B_t^x$ and using triangle inequality. The fourth relation is by noticing that $\overline{x}^{\leftarrow}_{-t_{\max}^2} + (2j-1)\delta^2 v^x_{-t_{\max}^2} + \left(B_{-(j-1)^2\delta^2}^x - B_{-t_{\max}^2}^x\right) = \overline{x}^{\leftarrow}_{-(j-1)^2\delta^2}$ and that $\overline{y}^{\leftarrow}_{-t_{\max}^2} + (2j-1)\delta^2 v^y_{-t_{\max}^2} + \left(B_{-(j-1)^2\delta^2}^x - B_t^x + B_t^y - B_{-t_{\max}^2}^y\right) \stackrel{d}{=} \overline{y}^{\leftarrow}_{-(j-1)^2\delta^2}$, and then using our assumption in the theorem statement that all processes are supported on a ball of radius $B/2$.

We now define a process $s_t$ defined by $ds_t = 2L_1/t_{\min}dt + 2dB_t^1$, initialized at $s_{-t_{\max}^2} = B \geq r_{-t_{\max}^2}$. We can verify that, up to time $\tau$, $r_t \leq s_t$ with probability 1. Let $\tau'$ denote the first-hitting time of $s_t$ to 0, then $\tau \leq \tau'$ with probability 1. Thus

$$
Pr(\tau \leq -t_{\min}^2) \geq Pr(\tau' \leq -t_{\min}^2) \geq 2Q\left(\frac{B}{2\sqrt{t_{\max}^2 - t_{\min}^2}}\right) \cdot e^{-BL_1/t_{\min} - L_1^2 t_{\max}^2/t_{\min}^2}
$$

where we apply Lemma 6. The proof follows by noticing that, if $\tau \leq -t_{\min}^2$, then $x_{t_{\min}} = y_{t_{\min}}$. This is because if $\tau \in [-k^2\delta^2, -(k-1)^2\delta^2]$, then $\overline{x}^{\leftarrow}_{-(k-1)^2\delta^2} = \overline{y}^{\leftarrow}_{-(k-1)^2\delta^2}$, and thus $\overline{x}_t^{\leftarrow} = \overline{y}_t^{\leftarrow}$ for all $t \geq -(k-1)^2\delta^2$, in particular, at $t = -t_{\min}^2$. $\qquad\square$

**Lemma 6.** *Consider the stochastic process*

$$
dr_t = dB_t^1 + cdt.
$$

*Assume that $r_0 \leq B/2$. Let $\tau$ denote the hitting time for $r_t = 0$. Then for any $T \in \mathbb{R}^+$,*

$$
Pr(\tau \leq T) \geq 2Q\left(\frac{B}{2\sqrt{T}}\right) \cdot e^{-ac - \frac{c^2 T}{2}},
$$

*where $Q$ is the tail probability of a standard Gaussian defined in Definition 1.*

*Proof.* We will use he following facts in our proof:

1. For $x \sim \mathcal{N}(0, \sigma^2)$, $Pr(x > r) = \frac{1}{2}\left(1 - erf\left(\frac{r}{\sqrt{2}\sigma}\right)\right) = \frac{1}{2}erfc\left(\frac{r}{\sqrt{2}\sigma}\right)$.

2. $\int_0^T \frac{a\exp\left(-\frac{a^2}{2t}\right)}{\sqrt{2\pi t^3}} dt = erfc\left(\frac{a}{\sqrt{2T}}\right) = 2Pr\left(\mathcal{N}(0, T) > a\right) = 2Q\left(\frac{a}{\sqrt{T}}\right)$ by definition of $Q$.

Let $dr_t = dB_t^1 + cdt$, with $r_0 = a$. The density of the hitting time $\tau$ is given by

$$p(\tau = t) = f(a, c, t) = \frac{a \exp\left(-\frac{(a+ct)^2}{2t}\right)}{\sqrt{2\pi t^3}}. \tag{18}$$

(see e.g. [3]). From item 2 above,

$$\int_0^T f(a, 0, t)dt = 2Q\left(\frac{a}{\sqrt{T}}\right).$$

In the case of a general $c \neq 0$, we can bound $\frac{(a+ct)^2}{2t} = \frac{a^2}{2t} + ac + \frac{c^2 t}{2}$. Consequently,

$$f(a, c, t) \geq f(a, 0, t) \cdot e^{-ac - \frac{c^2 t}{2}}.$$

Therefore,

$$Pr(\tau \leq T) = \int_0^T f(a, c, t)dt \geq \int_0^T f(a, 0, t)dt e^{-c} = 2Q\left(\frac{B}{2\sqrt{T}}\right) \cdot e^{-ac - \frac{c^2 T}{2}}.$$

$\square$

## A.5 TV Overlap

**Definition 1.** *Let $x$ be sampled from standard normal distribution $\mathcal{N}(0, 1)$. We define the Gaussian tail probability $Q(a) := Pr(x \geq a)$.*

**Lemma 7.** *We verify that for any two random vectors $\xi_x \sim \mathcal{N}(\mathbf{0}, \sigma^2 \mathbf{I})$ and $\xi_y \sim \mathcal{N}(\mathbf{0}, \sigma^2 \mathbf{I})$, each belonging to $\mathbb{R}^d$, the total variation distance between $x' = x + \xi_x$ and $y' = y + \xi_y$ is given by*

$$TV(x', y') = 1 - 2Q(r) \leq 1 - \frac{2r}{r^2 + 1} \frac{1}{\sqrt{2\pi}} e^{-r^2/2},$$

*where $r = \frac{\|x - y\|}{2\sigma}$, and $Q(r) = Pr(\xi \geq r)$, when $\xi \sim \mathcal{N}(0, 1)$.*

*Proof.* Let $\gamma := \frac{x-y}{\|x-y\|}$. We decompose $x', y'$ into the subspace/orthogonal space defined by $\gamma$:

$$x' = x^\perp + \xi_x^\perp + x^\| + \xi_x^\|$$
$$y' = y^\perp + \xi_y^\perp + y^\| + \xi_y^\|$$

where we define

$$x^\| := \gamma\gamma^T x \qquad x^\perp := x - x^\|$$
$$y^\| := \gamma\gamma^T y \qquad y^\perp := y - y^\|$$
$$\xi_x^\| := \gamma\gamma^T \xi_x \qquad \xi_x^\perp := \xi_x - \xi_x^\|$$
$$\xi_y^\| := \gamma\gamma^T \xi_y \qquad \xi_y^\perp := \xi_y - \xi_y^\|$$

We verify the independence $\xi_x^\perp \perp\!\!\!\perp \xi_x^\|$ and $\xi_y^\perp \perp\!\!\!\perp \xi_y^\|$ as they are orthogonal decompositions of the standard Gaussian. We will define a coupling between $x'$ and $y'$ by setting $\xi_x^\perp = \xi_y^\perp$. Under this coupling, we verify that

$$\left(x^\perp + \xi_x^\perp\right) - \left(y^\perp + \xi_y^\perp\right) = x - y - \gamma\gamma^T(x - y) = 0$$

Therefore, $x' = y'$ if and only if $x^\| + \xi_x^\| = y^\| + \xi_y^\|$. Next, we draw $(a, b)$ from the optimal coupling between $\mathcal{N}(0, 1)$ and $\mathcal{N}(\frac{\|x-y\|}{\sigma}, 1)$. We verify that $x^\| + \xi_x^\|$ and $y^\| + \xi_y^\|$ both lie in the span of $\gamma$. Thus it suffices to compare $\left\langle \gamma, x^\| + \xi_x^\| \right\rangle$ and $\left\langle \gamma, y^\| + \xi_y^\| \right\rangle$. We verify that $\left\langle \gamma, x^\| + \xi_x^\| \right\rangle =$

$\langle \gamma, y^{\parallel} \rangle + \langle \gamma, x^{\parallel} - y^{\parallel} \rangle + \langle \gamma, \xi_x^{\parallel} \rangle \sim \mathcal{N}(\langle \gamma, y^{\parallel} \rangle + \|x - y\|, \sigma^2) \stackrel{d}{=} \langle \gamma, y^{\parallel} \rangle + \sigma b$. We similarly verify that $\langle \gamma, y^{\parallel} + \xi_y^{\parallel} \rangle = \langle \gamma, y^{\parallel} \rangle + \langle \gamma, \xi_y^{\parallel} \rangle \sim \mathcal{N}(\langle \gamma, y^{\parallel} \rangle, \sigma^2) \stackrel{d}{=} \langle \gamma, y^{\parallel} \rangle + \sigma a$.

Thus $TV(x', y') = TV(\sigma a, \sigma b) = 1 - 2Q\left(\frac{\|x-y\|}{2\sigma}\right)$. The last inequality follows from

$$Pr(\mathcal{N}(0,1) \geq r) \geq \frac{r}{r^2 + 1} \frac{1}{\sqrt{2\pi}} e^{-r^2/2}$$

$\square$

## B    More on Restart Algorithm

### B.1    EDM Discretization Scheme

[13] proposes a discretization scheme for ODE given the starting $t_{\max}$ and end time $t_{\min}$. Denote the number of steps as $N$, then the EDM discretization scheme is:

$$t_{i<N} = \left(t_{\max}^{\frac{1}{\rho}} + \frac{i}{N-1}(t_{\min}^{\frac{1}{\rho}} - t_{\max}^{\frac{1}{\rho}})\right)^{\rho}$$

with $t_0 = t_{\max}$ and $t_{N-1} = t_{\min}$. $\rho$ is a hyperparameter that determines the extent to which steps near $t_{\min}$ are shortened. We adopt the value $\rho = 7$ suggested by [13] in all of our experiments. We apply the EDM scheme to creates a time discretization in each Restart interval $[t_{\max}, t_{\min}]$ in the Restart backward process, as well as the main backward process between $[0, T]$ (by additionally setting $t_{\min} = 0.002$ and $t_N = 0$ as in [13]). It is important to note that $t_{\min}$ should be included within the list of time steps in the main backward process to seamlessly incorporate the Restart interval into the main backward process. We summarize the scheme as a function in Algorithm 1.

---

**Algorithm 1** EDM_Scheme($t_{\min}, t_{\max}, N, \rho = 7$)

1: **return** $\left\{ (t_{\max}^{\frac{1}{\rho}} + \frac{i}{N-1}(t_{\min}^{\frac{1}{\rho}} - t_{\max}^{\frac{1}{\rho}}))^{\rho} \right\}_{i=0}^{N-1}$

---

### B.2    Restart Algorithm

We present the pseudocode for the Restart algorithm in Algorithm 2. In this pseudocode, we describe a more general case that applies $l$-level Restarting strategy. For each Restart segment, the include the number of steps in the Restart backward process $N_{\text{Restart}}$, the Restart interval $[t_{\min}, t_{\max}]$ and the number of Restart iteration $K$. We further denote the number of steps in the main backward process as $N_{\text{main}}$. We use the EDM discretization scheme (Algorithm 1) to construct time steps for the main backward process ($t_0 = T, t_{N_{\text{main}}} = 0$) as well as the Restart backward process, when given the starting/end time and the number of steps.

Although Heun's 2$^{\text{nd}}$ order method [2] (Algorithm 3) is the default ODE solver in the pseudocode, it can be substituted with other ODE solvers, such as Euler's method or the DPM solver [16].

The provided pseudocode in Algorithm 2 is tailored specifically for diffusion models [13]. To adapt Restart for other generative models like PFGM++ [28], we only need to modify the Gaussian perturbation kernel in the Restart forward process (line 10 in Algorithm 2) to the one used in PFGM++.

## C    Experimental Details

In this section, we discuss the configurations for different samplers in details. All the experiments are conducted on eight NVIDIA A100 GPUs.

---

**Algorithm 2** Restart sampling

---

1: **Input:** Score network $s_\theta$, time steps in main backward process $t_{i \in \{0, N_{\text{main}}\}}$, Restart parameters $\{(N_{\text{Restart},j}, K_j, t_{\min,j}, t_{\max,j})\}_{j=1}^l$
2: Round $t_{\min,j \in \{1,l\}}$ to its nearest neighbor in $t_{i \in \{0, N_{\text{main}}\}}$
3: Sample $x_0 \sim \mathcal{N}(\mathbf{0}, T^2 \boldsymbol{I})$
4: **for** $i = 0 \ldots N_{\text{main}} - 1$ **do**                          ▷ Main backward process
5:     $x_{t_{i+1}} = \text{OneStep\_Heun}(s_\theta, t_i, t_{i+1})$               ▷ Running single step ODE
6:     **if** $\exists j \in \{1, \ldots, l\}, t_{i+1} = t_{\min,j}$ **then**
7:         $t_{\min} = t_{\min,j}, t_{\max} = t_{\max,j}, K = K_j, N_{\text{Restart}} = N_{\text{Restart},j}$
8:         $x_{t_{\min}}^0 = x_{t_{i+1}}$
9:         **for** $k = 0 \ldots K - 1$ **do**                          ▷ **Restart for $K$ iterations**
10:             $\varepsilon_{t_{\min} \to t_{\max}} \sim \mathcal{N}(\mathbf{0}, (t_{\max}^2 - t_{\min}^2)\boldsymbol{I})$
11:             $x_{t_{\max}}^{k+1} = x_{t_{\min}}^k + \varepsilon_{t_{\min} \to t_{\max}}$               ▷ **Restart forward process**
12:             $\{\bar{t}_m\}_{m=0}^{N_{\text{Restart}}-1} = \text{EDM\_Scheme}(t_{\min}, t_{\max}, N_{\text{Restart}})$
13:             **for** $m = 0 \ldots N_{\text{Restart}} - 1$ **do**               ▷ **Restart backward process**
14:                 $x_{\bar{t}_{m+1}}^{k+1} = \text{OneStep\_Heun}(s_\theta, \bar{t}_m, \bar{t}_{m+1})$
15:             **end for**
16:         **end for**
17:     **end if**
18: **end for**
19: **return** $x_{t_{N_{\text{main}}}}$

---

---

**Algorithm 3** OneStep\_Heun($s_\theta, x_{t_i}, t_i, t_{i+1}$)

---

1: $d_i = t_i s_\theta(x_{t_i}, t_i)$
2: $x_{t_{i+1}} = x_{t_i} - (t_{i+1} - t_i)d_i$
3: **if** $t_{i+1} \neq 0$ **then**
4:     $d_i' = t_{i+1} s_\theta(x_{t_{i+1}}, t_{i+1})$
5:     $x_{t_{i+1}} = x_{t_i} - (t_{i+1} - t_i)(\frac{1}{2}d_i + \frac{1}{2}d_i')$
6: **end if**
7: **return** $x_{t_{i+1}}$

---

### C.1 Configurations for Baselines

We select **Vanilla SDE** [23], **Improved SDE** [13], **Gonna Go Fast** [12] as SDE baselines and the **Heun**'s 2nd order method [2] (Alg 3) as ODE baseline on standard benchmarks CIFAR-10 and ImageNet $64 \times 64$. We choose **DDIM** [22], **Heun**'s 2nd order method, and **DDPM** [9] for comparison on Stable Diffusion model.

Vanilla SDE denotes the reverse-time SDE sampler in [23]. For Improved SDE, we use the recommended dataset-specific hyperparameters (*e.g.*, $S_{\max}, S_{\min}, S_{\text{churn}}$) in Table 5 of the EDM paper [13]. They obtained these hyperparameters by grid search. Gonna Go Fast [12] applied an adaptive step size technique based on Vanilla SDE and we directly report the FID scores listed in [12] for Gonna Go Fast on CIFAR-10 (VP). For fair comparison, we use the EDM discretization scheme [13] for Vanilla SDE, Improved SDE, Heun as well as Restart.

We borrow the hyperparameters such as discretization scheme or initial noise scale on Stable Diffusion models in the diffuser [3] code repository. We directly use the DDIM and DDPM samplers implemented in the repo. We apply the same set of hyperparameters to Heun and Restart.

### C.2 Configurations for Restart

We report the configurations for Restart for different models and NFE on standard benchmarks CIFAR-10 and ImageNet $64 \times 64$. The hyperparameters of Restart include the number of steps in the main backward process $N_{\text{main}}$, the number of steps in the Restart backward process $N_{\text{Restart}}$, the Restart interval $[t_{\min}, t_{\max}]$ and the number of Restart iteration $K$. In Table 3 (CIFAR-10, VP)

---

[3]https://github.com/huggingface/diffusers

we provide the quintuplet $(N_{\text{main}}, N_{\text{Restart}}, t_{\min}, t_{\max}, K)$ for each experiment. Since we apply the multi-level Restart strategy for ImageNet $64 \times 64$, we provide $N_{\text{main}}$ as well as a list of quadruple $\{(N_{\text{Restart},i}, K_i, t_{\min,i}, t_{\max,i})\}_{i=1}^l$ ($l$ is the number of Restart interval depending on experiments) in Table 5. In order to integrate the Restart time interval to the main backward process, we round $t_{\min,i}$ to its nearest neighbor in the time steps of main backward process, as shown in line 2 of Algorithm 2. We apply Heun method for both main/backward process. The formula for NFE calculation is NFE $= \underbrace{2 \cdot N_{\text{main}} - 1}_{\text{main backward process}} + \sum_{i=1}^l \underbrace{K_i}_{\text{number of repetitions}} \cdot \underbrace{(2 \cdot (N_{\text{Restart},i} - 1))}_{\text{per iteration in } i^{\text{th}} \text{ Restart interval}}$ in this case. Inspired by [13], we inflate the additive noise in the Restart forward process by multiplying $S_{\text{noise}} = 1.003$ on ImageNet $64 \times 64$, to counteract the over-denoising tendency of neural networks. We also observe that setting $\gamma = 0.05$ in Algorithm 2 of EDM [13] would sligtly boost the Restart performance on ImageNet $64 \times 64$ when $t \in [0.01, 1]$.

We further include the configurations for Restart on Stable Diffusion models in Table 10, with a varying guidance weight $w$. Similar to ImageNet $64 \times 64$, we use multi-level Restart with a fixed number of steps $N_{\text{main}} = 30$ in the main backward process. We utilize the Euler method for the main backward process and the Heun method for the Restart backward process, as our empirical observations indicate that the Heun method doesn't yield significant improvements over the Euler method, yet necessitates double the steps. The number of steps equals to $N_{\text{main}} + \sum_{i=1}^l K_i \cdot (2 \cdot (N_{\text{Restart},i} - 1))$ in this case. We set the total number of steps to 66, including main backward process and Restart backward process.

Given the prohibitively large search space for each Restart quadruple, a comprehensive enumeration of all possibilities is impractical due to computational limitations. Instead, we adjust the configuration manually, guided by the heuristic that weaker/smaller models or more challenging tasks necessitate a stronger Restart strength (e.g., larger $K$, wider Restart interval, etc). On average, we select the best configuration from 5 sets for each experiment; these few trials have empirically outperformed previous SDE/ODE samplers. We believe that developing a systematic approach for determining Restart configurations could be of significant value in the future.

### C.3   Pre-trained Models

For CIFAR-10 dataset, we use the pre-trained VP and EDM models from the EDM repository [4], and PFGM++ ($D = 2048$) model from the PFGM++ repository [5]. For ImageNet $64 \times 64$, we borrow the pre-trained EDM model from EDM repository as well.

### C.4   Classifier-free Guidance

We follow the convention in [20], where each step in classifier-free guidance is as follows:

$$\tilde{s}_\theta(x, c, t) = w s_\theta(x, c, t) + (1 - w) s_\theta(x, t)$$

where $c$ is the conditions, and $s_\theta(x, c, t)/s_\theta(x, t)$ is the conditional/unconditional models, sharing parameters. Increasing $w$ would strength the effect of guidance, usually leading to a better text-image alignment [20].

### C.5   More on the Synthetic Experiment

### C.5.1   Discrete Dataset

We generate the underlying discrete dataset $S$ with $|S| = 2000$ as follows. Firstly, we sample 2000 points, denoted as $S_1$, from a mixture of two Gaussians in $\mathbb{R}^4$. Next, we project these points onto $\mathbb{R}^{20}$. To ensure a variance of 1 on each dimension, we scale the coordinates accordingly. This setup aims to simulate data points that primarily reside on a lower-dimensional manifold with multiple modes.

The specific details are as follows: $S_1 \sim 0.3N(a, s^2 I) + 0.7(-a, s^2 I)$, where $a = (3, 3, 3, 3) \subset \mathbb{R}^4$ and $s = 1$. Then, we randomly select a projection matrix $P \in \mathbb{R}^{20 \times 4}$, where each entry is drawn from $N(0, 1)$, and compute $S_2 = PS_1$. Finally, we scale each coordinate by a constant factor to ensure a variance of 1.

---

[4] https://github.com/NVlabs/edm
[5] https://github.com/Newbeeer/pfgmpp

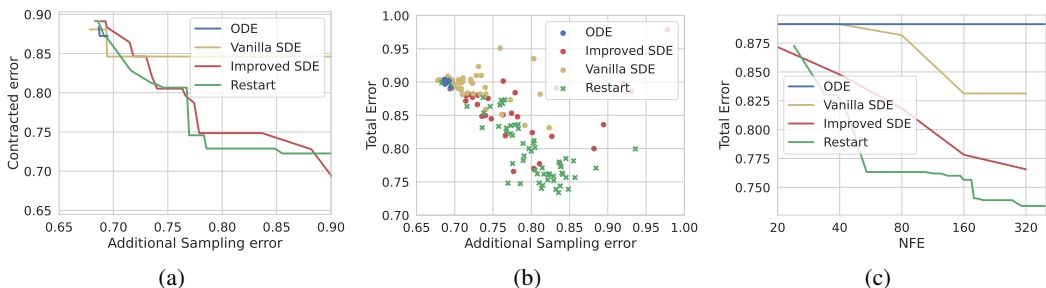

Figure 8: Comparison of additional sampling error versus **(a)** contracted error (plotting the Pareto frontier) and **(b)** total error (using a scatter plot). **(c)** Pareto frontier of NFE versus total error.

### C.5.2  Model Architecture

We employ a common MLP architecture with a latent size of 64 to learn the score function. The training method is adapted from [13], which includes the preconditioning technique and denoising score-matching objective [25].

### C.5.3  Varying Hyperparameters

To achieve the best trade-off between contracted error and additional sampling error, and optimize the NFE versus FID (Fréchet Inception Distance) performance, we explore various hyperparameters. [13] shows that the Vanilla SDE can be endowed with additional flexibility by varying the coefficient $\beta(t)$ (Eq.(6) in [13]). Hence, regarding SDE, we consider NFE values from $\{20, 40, 80, 160, 320\}$, and multiply the original $\beta(t) = \dot{\sigma}(t)/\sigma(t)$ [13] with values from $\{0, 0.25, 0.5, 1, 1.5, 2, 4, 8\}$. It is important to note that larger NFE values do not lead to further performance improvements. For restarts, we tried the following two settings: first we set the number of steps in Restart backward process to 40 and vary the number of Restart iterations $K$ in the range $\{0, 5, 10, 15, 20, 25, 30, 35\}$. We also conduct a grid search with the number of Restart iterations $K$ ranging from 5 to 25 and the number of steps in Restart backward process varying from 2 to 7. For ODE, we experiment with the number of steps set to $\{20, 40, 80, 160, 320, 640\}$.

Additionally, we conduct an experiment for Improved SDE in EDM. We try different values of $S_{\text{churn}}$ in the range of $\{0, 1, 2, 4, 8, 16, 32, 48, 64\}$. We also perform a grid search where the number of steps ranged from 20 to 320 and $S_{\text{churn}}$ takes values of $[0.2 \times \text{steps}, 0.5 \times \text{steps}, 20, 60]$. The plot combines the results from SDE and is displayed in Figure 8.

To mitigate the impact of randomness, we collect the data by averaging the results from five runs with the same hyperparameters. To compute the Wasserstein distance between two discrete distributions, we use minimum weight matching.

### C.5.4  Plotting the Pareto frontier

We generate the Pareto frontier plots as follows. For the additional sampling error versus contracted error plot, we first sort all the data points based on their additional sampling error and then connect the data points that represent prefix minimums of the contracted error. Similarly, for the NFE versus FID plot, we sort the data points based on their NFE values and connect the points where the FID is a prefix minimum.

## D  Extra Experimental Results

### D.1  Numerical Results

In this section, we provide the corresponding numerical reuslts of Fig. 3(a) and Fig. 3(b), in Table 2, 3 (CIFAR-10 VP, EDM, PFGM++) and Table 4, 5 (ImageNet $64 \times 64$ EDM), respectively. We also include the performance of Vanilla SDE in those tables. For the evaluation, we compute the Fréchet distance between 50000 generated samples and the pre-computed statistics of CIFAR-10 and ImageNet $64 \times 64$. We follow the evaluation protocol in EDM [13] that calculates each FID scores three times with different seeds and report the minimum.

We also provide the numerical results on the Stable Diffusion model [19], with a classifier guidance weight $w = 2, 3, 5, 8$ in Table 6, 7, 8, 9. As in [17], we report the zero-shot FID score on 5K random prompts sampled from the COCO validation set. We evaluate CLIP score [6] with the open-sourced ViT-g/14 [11], Aesthetic score by the more recent LAION-Aesthetics Predictor V2 [6]. We average the CLIP and Aesthetic scores over 5K generated samples. The number of function evaluations is two times the sampling steps in Stable Diffusion model, since each sampling step involves the evaluation of the conditional and unconditional model.

Table 2: CIFAR-10 sample quality (FID score) and number of function evaluations (NFE) on VP [23] for baselines

|  | NFE | FID |
|---|---|---|
| *ODE (Heun)* [13] | 1023 | 2.90 |
|  | 511 | 2.90 |
|  | 255 | 2.90 |
|  | 127 | 2.90 |
|  | 63 | 2.89 |
|  | 35 | 2.97 |
| *Vanilla SDE* [23] | 1024 | 2.79 |
|  | 512 | 4.01 |
|  | 256 | 4.79 |
|  | 128 | 12.57 |
| *Gonna Go Fast* [12] | 1000 | 2.55 |
|  | 329 | 2.70 |
|  | 274 | 2.74 |
|  | 179 | 2.59 |
|  | 147 | 2.95 |
|  | 49 | 72.29 |
| *Improved SDE* [13] | 1023 | 2.35 |
|  | 511 | 2.37 |
|  | 255 | 2.40 |
|  | 127 | 2.58 |
|  | 63 | 2.88 |
|  | 35 | 3.45 |

Table 3: CIFAR-10 sample quality (FID score), number of function evaluations (NFE) and Restart configurations on VP [23], EDM [13] and PFGM++ [28]

| Method | NFE | FID | Configuration $(N_{\mathrm{main}}, N_{\mathrm{Restart},i}, K_i, t_{\mathrm{min},i}, t_{\mathrm{max},i})$ |
|---|---|---|---|
| *VP* |  |  |  |
|  | 519 | 2.11 | $(20, 9, 30, 0.06, 0.20)$ |
|  | 115 | 2.21 | $(18, 3, 20, 0.06, 0.30)$ |
|  | 75 | 2.27 | $(18, 3, 10, 0.06, 0.30)$ |
|  | 55 | 2.45 | $(18, 3, 5, 0.06, 0.30)$ |
|  | 43 | 2.70 | $(18, 3, 2, 0.06, 0.30)$ |
| *EDM* |  |  |  |
|  | 43 | 1.90 | $(18, 3, 2, 0.14, 0.30)$ |
| *PFGM++* |  |  |  |
|  | 43 | 1.88 | $(18, 3, 2, 0.14, 0.30)$ |

---

[6]https://github.com/christophschuhmann/improved-aesthetic-predictor

Table 4: ImageNet $64 \times 64$ sample quality (FID score) and number of function evaluations (NFE) on EDM [13] for baselines

|  | NFE | FID (50k) |
|---|---|---|
| *ODE (Heun)* [13] | 1023 | 2.24 |
|  | 511 | 2.24 |
|  | 255 | 2.24 |
|  | 127 | 2.25 |
|  | 63 | 2.30 |
|  | 35 | 2.46 |
| *Vanilla SDE* [23] | 1024 | 1.89 |
|  | 512 | 3.38 |
|  | 256 | 11.91 |
|  | 128 | 59.71 |
| *Improved SDE* [13] | 1023 | 1.40 |
|  | 511 | 1.45 |
|  | 255 | 1.50 |
|  | 127 | 1.75 |
|  | 63 | 2.24 |
|  | 35 | 2.97 |

Table 5: ImageNet $64 \times 64$ sample quality (FID score), number of function evaluations (NFE) and Restart configurations on EDM [13]

| NFE | FID (50k) | Configuration $N_{\text{main}}, \{(N_{\text{Restart},i}, K_i, t_{\text{min},i}, t_{\text{max},i})\}_{i=1}^{l}$ |
|---|---|---|
| 623 | 1.36 | 36, {(10, 3, 19.35, 40.79),(10, 3, 1.09, 1.92), (7, 6, 0.59, 1.09), (7, 6, 0.30, 0.59), (7, 25, 0.06, 0.30)} |
| 535 | 1.39 | 36, {(6, 1, 19.35, 40.79),(6, 1, 1.09, 1.92), (7, 6, 0.59, 1.09), (7, 6, 0.30, 0.59), (7, 25, 0.06, 0.30)} |
| 385 | 1.41 | 36, {(3, 1, 19.35, 40.79),(6, 1, 1.09, 1.92), (6, 5, 0.59, 1.09), (6, 5, 0.30, 0.59), (6, 20, 0.06, 0.30)} |
| 203 | 1.46 | 36, {(4, 1, 19.35, 40.79),(4, 1, 1.09, 1.92), (4, 5, 0.59, 1.09), (4, 5, 0.30, 0.59), (6, 6, 0.06, 0.30)} |
| 165 | 1.51 | 18, {(3, 1, 19.35, 40.79),(4, 1, 1.09, 1.92), (4, 5, 0.59, 1.09), (4, 5, 0.30, 0.59), (4, 10, 0.06, 0.30)} |
| 99 | 1.71 | 18, {(3, 1, 19.35, 40.79),(4, 1, 1.09, 1.92), (4, 4, 0.59, 1.09), (4, 1, 0.30, 0.59), (4, 4, 0.06, 0.30)} |
| 67 | 1.95 | 18, {(5, 1, 19.35, 40.79),(5, 1, 1.09, 1.92), (5, 1, 0.59, 1.09), (5, 1, 0.06, 0.30)} |
| 39 | 2.38 | 14, {(3, 1, 19.35, 40.79), (3, 1, 1.09, 1.92), (3, 1, 0.06, 0.30)} |

Table 6: Numerical results on Stable Diffusion v1.5 with a classifier-free guidance weight $w = 2$

|  | Steps | FID (5k) ↓ | CLIP score ↑ | Aesthetic score ↑ |
|---|---|---|---|---|
| *DDIM* [22] | 50 | 16.08 | 0.2905 | 5.13 |
|  | 100 | 15.35 | 0.2920 | 5.15 |
| *Heun* | 51 | 18.80 | 0.2865 | 5.14 |
|  | 101 | 18.21 | 0.2871 | 5.15 |
| *DDPM* [9] | 100 | 13.53 | 0.3012 | 5.20 |
|  | 200 | 13.22 | 0.2999 | 5.19 |
| *Restart* | 66 | 13.16 | 0.2987 | 5.19 |

Table 7: Numerical results on Stable Diffusion v1.5 with a classifier-free guidance weight $w = 3$

|  | Steps | FID (5k) ↓ | CLIP score ↑ | Aesthetic score ↑ |
|---|---|---|---|---|
| *DDIM* [22] | 50 | 14.28 | 0.3056 | 5.22 |
|  | 100 | 14.30 | 0.3056 | 5.22 |
| *Heun* | 51 | 15.63 | 0.3022 | 5.20 |
|  | 101 | 15.40 | 0.3026 | 5.21 |
| *DDPM* [9] | 100 | 15.72 | 0.3129 | 5.28 |
|  | 200 | 15.13 | 0.3131 | 5.28 |
| *Restart* | 66 | 14.48 | 0.3079 | 5.25 |

Table 8: Numerical results on Stable Diffusion v1.5 with a classifier-free guidance weight $w = 5$

|  | Steps | FID (5k) ↓ | CLIP score ↑ | Aesthetic score ↑ |
|---|---|---|---|---|
| *DDIM* [22] | 50 | 16.60 | 0.3154 | 5.31 |
|  | 100 | 16.80 | 0.3157 | 5.31 |
| *Heun* | 51 | 16.26 | 0.3135 | 5.28 |
|  | 101 | 16.38 | 0.3136 | 5.29 |
| *DDPM* [9] | 100 | 19.62 | 0.3197 | 5.36 |
|  | 200 | 18.88 | 0.3200 | 5.35 |
| *Restart* | 66 | 16.21 | 0.3179 | 5.33 |

Table 9: Numerical results on Stable Diffusion v1.5 with a classifier-free guidance weight $w = 8$

|  | Steps | FID (5k) ↓ | CLIP score ↑ | Aesthetic score ↑ |
|---|---|---|---|---|
| *DDIM* [22] | 50 | 19.83 | 0.3206 | 5.37 |
|  | 100 | 19.82 | 0.3200 | 5.37 |
| *Heun* | 51 | 18.44 | 0.3186 | 5.35 |
|  | 101 | 18.72 | 0.3185 | 5.36 |
| *DDPM* [9] | 100 | 22.58 | 0.3223 | 5.39 |
|  | 200 | 21.67 | 0.3212 | 5.38 |
| *Restart* | 47 | 18.40 | 0.3228 | 5.41 |

## D.2 Sensitivity Analysis of Hyper-parameters

We also investigate the impact of varying $t_{\min}$ when $t_{\max} = t_{\min} + 0.3$, and the length the restart interval when $t_{\min} = 0.06$. Fig. 10(a) reveals that FID scores achieve a minimum at a $t_{\min}$ close to 0

Table 10: Restart (Steps=66) configurations on Stable Diffusion v1.5

| $w$ | Configuration $N_{\text{main}}, \{(N_{\text{Restart},i}, K_i, t_{\min,i}, t_{\max,i})\}_{i=1}^{l}$ |
|---|---|
| 2 | 30, {(5, 2, 1, 9), (5, 2, 5, 10)} |
| 3 | 30, {(10, 2, 0.1, 3)} |
| 5 | 30, {(10, 2 0.1, 2)} |
| 8 | 30, {(10, 2, 0.1, 2)} |

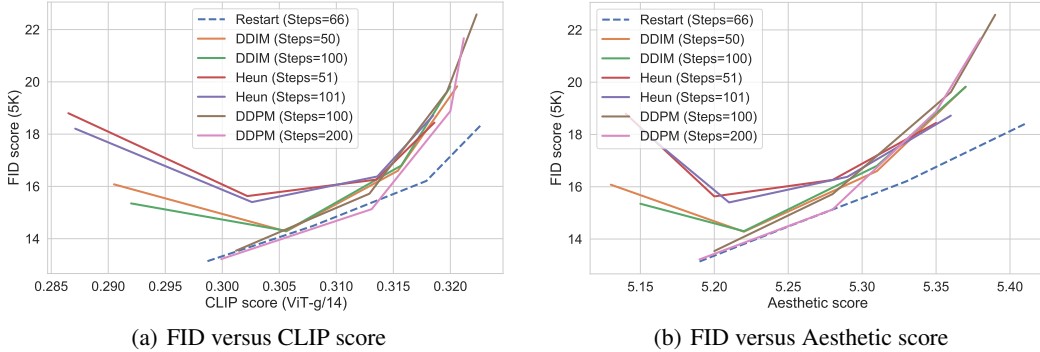

(a) FID versus CLIP score      (b) FID versus Aesthetic score

Figure 9: FID score versus **(a)** CLIP ViT-g/14 score and **(b)** Aesthetic score for text-to-image generation at $512 \times 512$ resolution, using Stable Diffusion v1.5 with varying classifier-free guidance weight $w = 2, 3, 5, 8$.

on VP, indicating higher accumulated errors at the end of sampling and poor neural estimations at small $t$. Note that the Restart interval $0.3$ is about twice the length of the one in Table 1 and Restart does not outperform the ODE baseline on EDM. This suggests that, as a rule of thumb, we should apply greater Restart strength (*e.g.*, larger $K$, $t_{\max} - t_{\min}$) for weaker or smaller architectures and vice versa.

In theory, a longer interval enhances contraction but may add more additional sampling errors. Again, the balance between these factors results in a V-shaped trend in our plots (Fig. 10(b)). In practice, selecting $t_{\max}$ close to the dataset's radius usually ensures effective mixing when $t_{\min}$ is small.

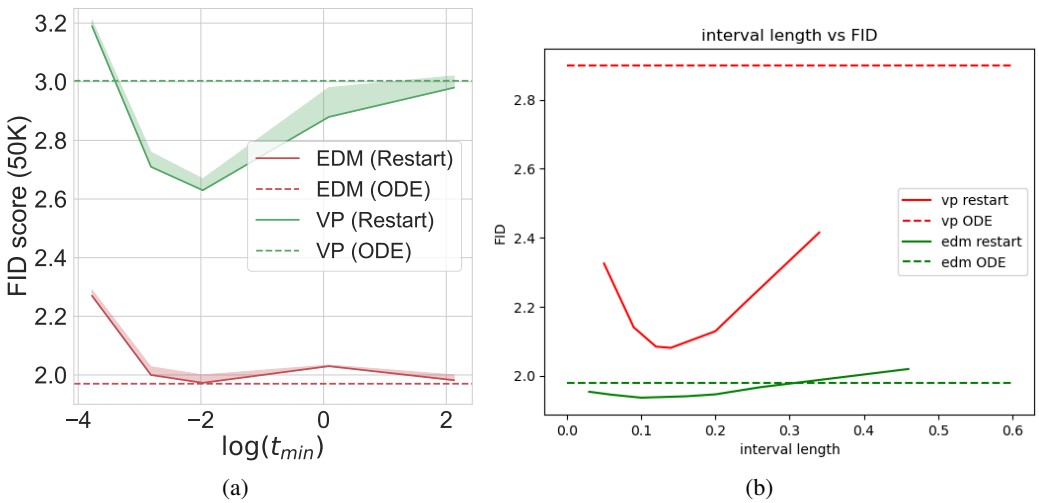

(a)          (b)

Figure 10: (a): Adjusting $t_{\min}$ in Restart on VP/EDM; (b): Adjusting the Restart interval length when $t_{\min} = 0.06$.

# E    Extended Generated Images

In this section, we provide extended generated images by Restart, DDIM, Heun and DDPM on text-to-image Stable Diffusion v1.5 model [19]. We showcase the samples of four sets of text prompts in Fig. 11, Fig. 12, Fig. 13, Fig. 14, with a classifier-guidance weight $w = 8$.

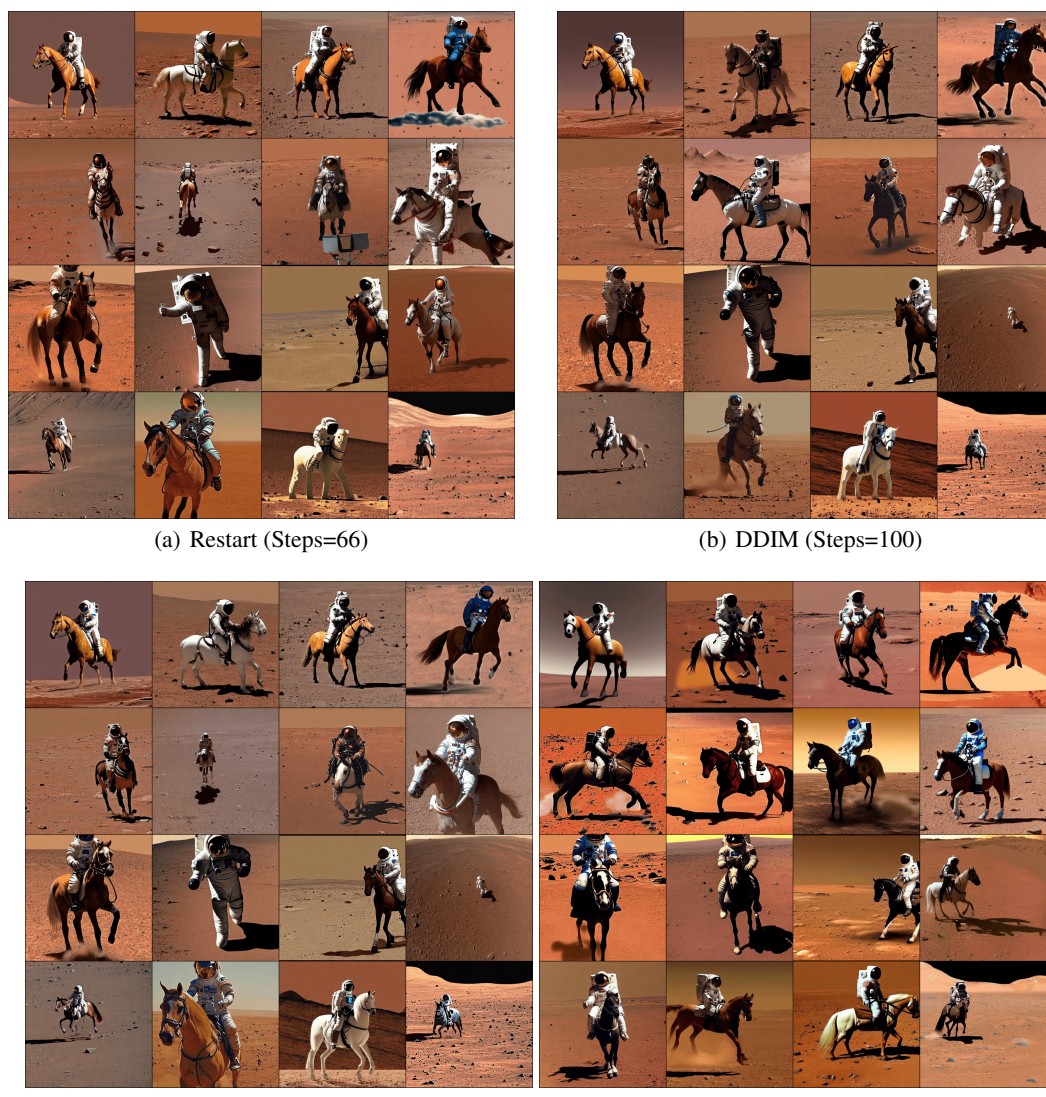

(a) Restart (Steps=66)    (b) DDIM (Steps=100)

(c) Heun (Steps=101)    (d) DDPM (Steps=100)

Figure 11: Generated images with text prompt="A photo of an astronaut riding a horse on mars" and $w = 8$.

# F    Heun's method is DPM-Solver-2 (with $r_2 = 1$)

The first order ODE in DPM-Solver [16] (DPM-Solver-1) is in the form of:

$$\hat{x}_{t_{i-1}} = \frac{\alpha_{t_i}}{\alpha_{t_{i-1}}}\hat{x}_{t_{i-1}} - (\hat{\sigma}_{t_{i-1}}\frac{\alpha_{t_i}}{\alpha_{t_{i-1}}} - \hat{\sigma}_{t_i})\hat{\sigma}_{t_i}\nabla_x \log p_{\hat{\sigma}_{t_i}}(\hat{x}_{t_i}) \tag{19}$$

The first order ODE in EDM is in the form of

$$x_{t_{i-1}} = x_{t_i} - (\sigma_{t_{i-1}} - \sigma_{t_i})\sigma_{t_i}\nabla_x \log p_{\sigma_{t_i}}(x_{t_i}) \tag{20}$$

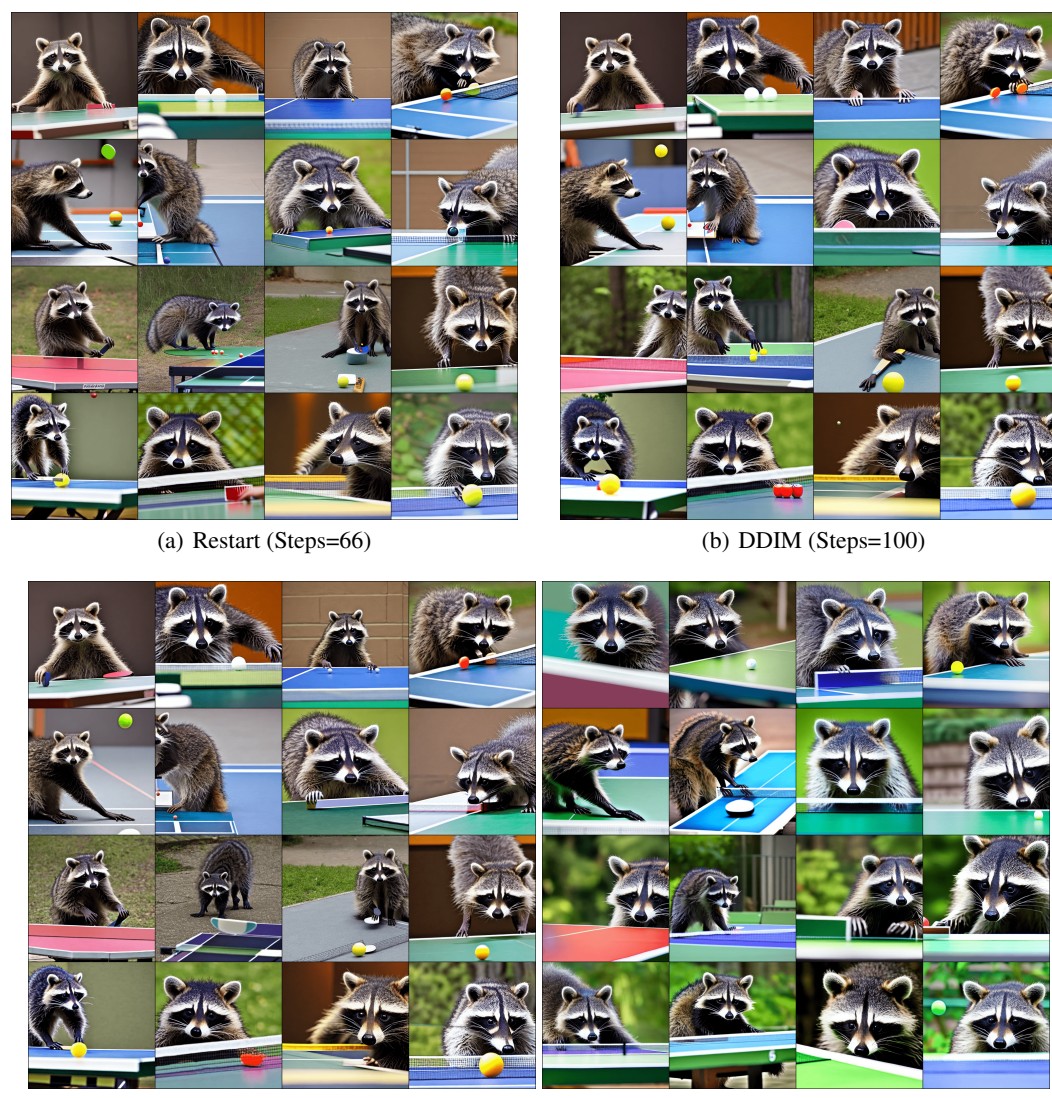

| | |
|---|---|
| (a) Restart (Steps=66) | (b) DDIM (Steps=100) |
| (c) Heun (Steps=101) | (d) DDPM (Steps=100) |

Figure 12: Generated images with text prompt="A raccoon playing table tennis" and $w = 8$.

When $x_t = \frac{\hat{x}_t}{\alpha_t}$, $\hat{\sigma}_t = \sigma_t \alpha_t$, we can rewrite the DPM-Solver-1 (Eq. (19)) as:

$$
\begin{aligned}
x_{t_{i-1}} &= x_{t_i} - (\sigma_{t_{i-1}} - \sigma_{t_i})\hat{\sigma}_{t_i} \nabla_x \log p_{\hat{\sigma}_{t_i}}(\hat{x}_{t_i}) \\
&= x_{t_i} - (\sigma_{t_{i-1}} - \sigma_{t_i})\hat{\sigma}_{t_i} \nabla_x \log p_{\sigma_{t_i}}(x_{t_i})\frac{1}{\alpha_{t_i}} \quad \text{(change-of-variable)} \\
&= x_{t_i} - (\sigma_{t_{i-1}} - \sigma_{t_i})\sigma_{t_i} \nabla_x \log p_{\sigma_{t_i}}(x_{t_i})
\end{aligned}
$$

where the expression is exact the same as the ODE in EDM [13]. It indicates that the sampling trajectory in DPM-Solver-1 is equivalent to the one in EDM, up to a time-dependent scaling ($\alpha_t$). As $\lim_{t \to 0} \alpha_t = 1$, the two solvers will leads to the same final points when using the same time discretization. Note that the DPM-Solver-1 is also equivalent to DDIM (*c.f.* Section 4.1 in [16]), as also used in this paper.

With that, we can further verify that the Heun's method used in this paper corresponds to the DPM-Solver-2 when setting $r_1 = 1$.

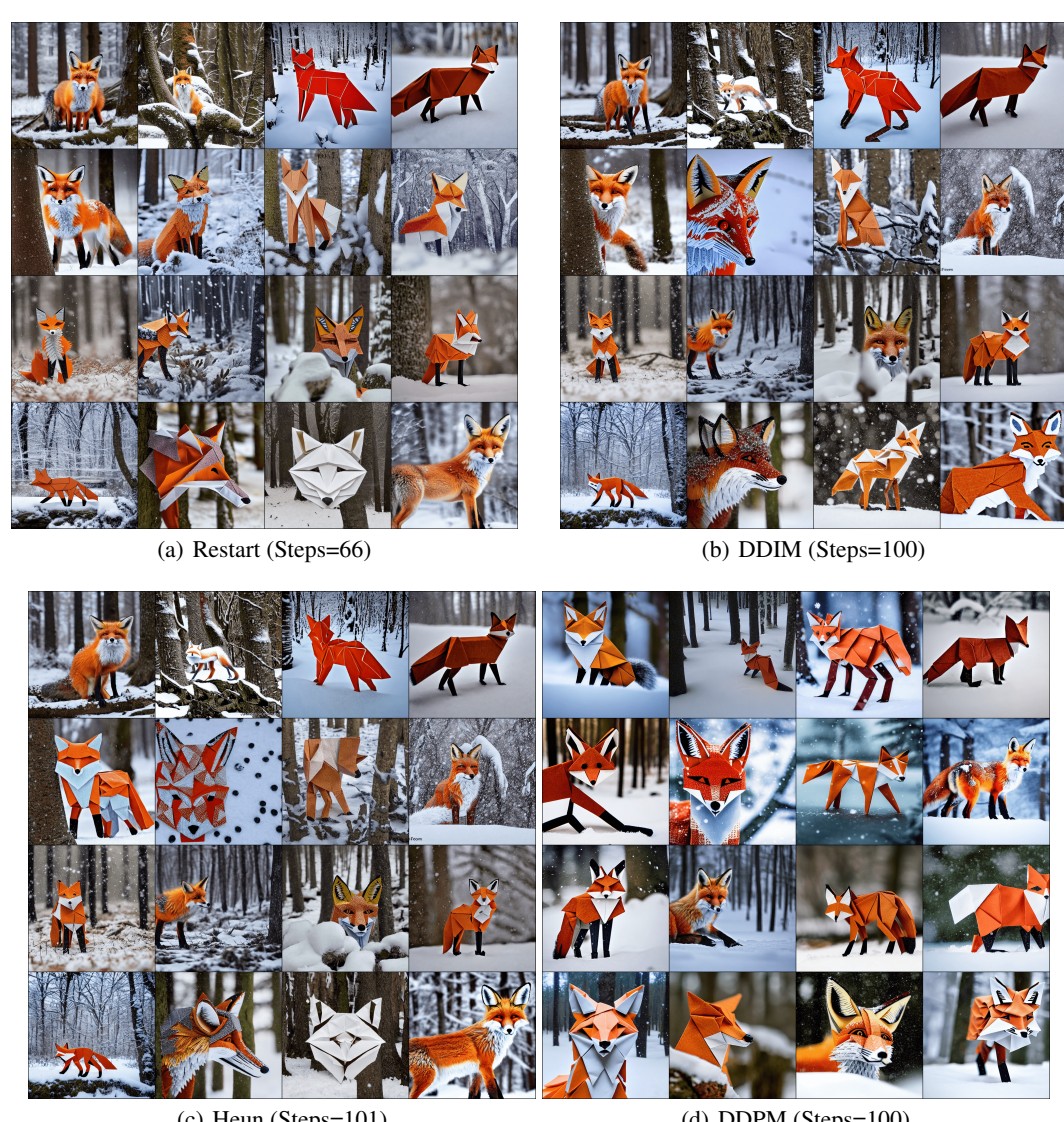

(a) Restart (Steps=66)

(b) DDIM (Steps=100)

(c) Heun (Steps=101)

(d) DDPM (Steps=100)

Figure 13: Generated images with text prompt="Intricate origami of a fox in a snowy forest" and $w = 8$.

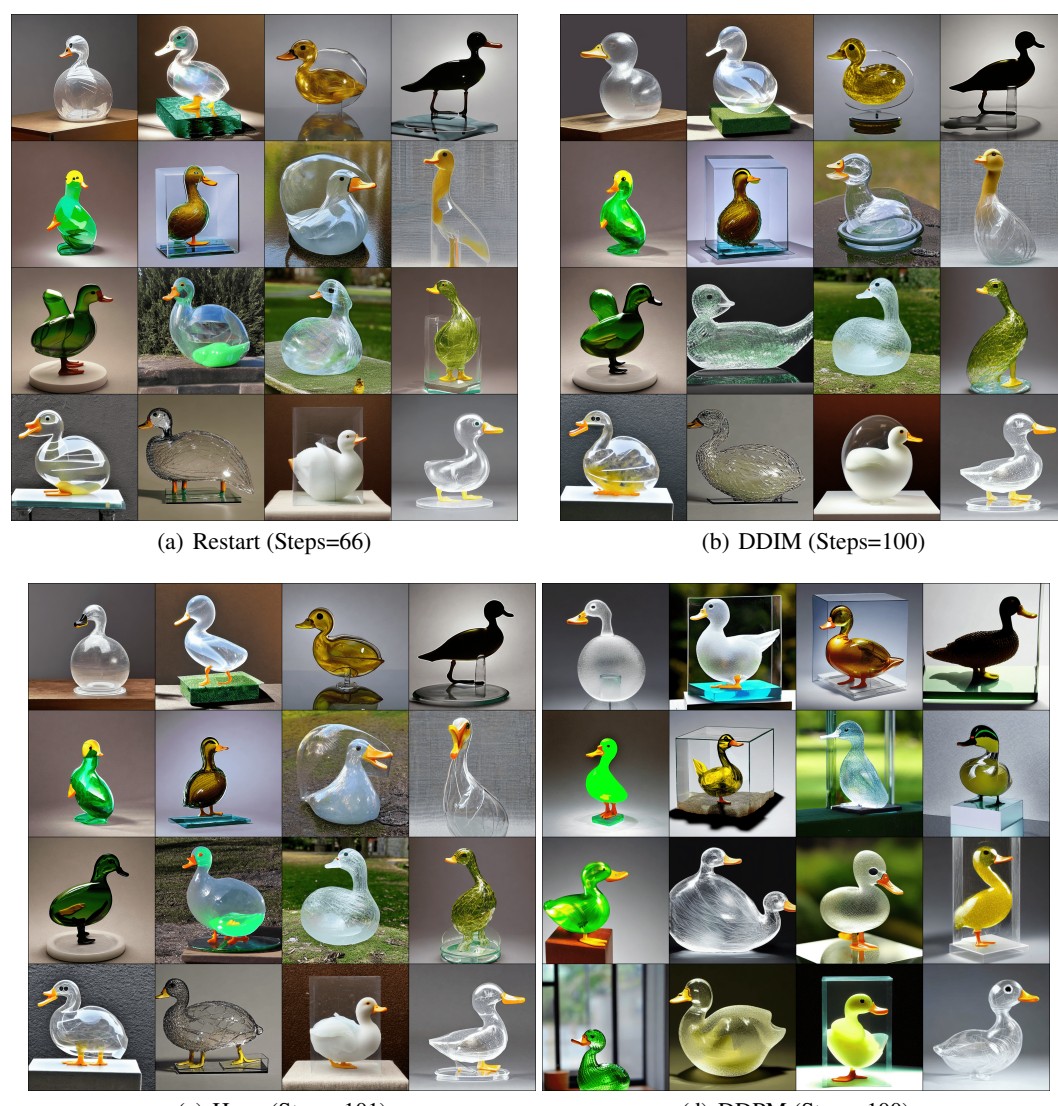

(a) Restart (Steps=66)

(b) DDIM (Steps=100)

(c) Heun (Steps=101)

(d) DDPM (Steps=100)

Figure 14: Generated images with text prompt="A transparent sculpture of a duck made out of glass" and $w = 8$.

# G Broader Impact

The field of deep generative models incorporating differential equations is rapidly evolving and holds significant potential to shape our society. Nowadays, a multitude of photo-realistic images generated by text-to-image Stable Diffusion models populate the internet. Our work introduces Restart, a novel sampling algorithm that outperforms previous samplers for diffusion models and PFGM++. With applications extending across diverse areas, the Restart sampling algorithm is especially suitable for generation tasks demanding high quality and rapid speed. Yet, it is crucial to recognize that the utilization of such algorithms can yield both positive and negative repercussions, contingent on their specific applications. On the one hand, Restart sampling can facilitate the generation of highly realistic images and audio samples, potentially advancing sectors such as entertainment, advertising, and education. On the other hand, it could also be misused in *deepfake* technology, potentially leading to social scams and misinformation. In light of these potential risks, further research is required to develop robustness guarantees for generative models, ensuring their use aligns with ethical guidelines and societal interests.

