# OpenReview forum: "Restart Sampling for Improving Generative Processes"
_NeurIPS.cc/2023/Conference — NeurIPS 2023 poster_

### Official Review · Reviewer_kog8 · 2023-07-04

**Soundness:** 3 good
**Presentation:** 3 good
**Contribution:** 3 good
**Rating:** 5
**Confidence:** 4

**Summary:**

This paper proposes analyzes SDE and ODE-based samplers for diffusion models. Based on the analysis, this paper introduces a new solver, Restart, for sampling from diffusion models. The effectiveness of Restart is demonstrated on various unconditional and conditional generation tasks.

**Strengths:**

- The paper is well-written.
- The proposed method shows better performance than EDM at moderate NFE regions.

**Weaknesses:**

I am willing to raise the score by 1 or 2 points if the authors address my concerns satisfactorily.

**Weakness 1 : Ambiguity regarding Theorems 1 and 2.**
- For Theorem 1, if we set $[t_{\min},t_{\max}] = [0,T]$, the terms for contracted errors $TV(p_T^{ODE_\theta},p_T)$ and $TV(p_T^{SDE_\theta},p_T)$ vanish because $p_T^{ODE_\theta}$, $p_T^{SDE_\theta}$, and $p_T$ are all identically Gaussian distributions. Then, we end up with terms depending on $\delta$, $\epsilon_{approx}$, and $t_{\max} - t_{\min}$ only, so Theorem 1 does not provide any insight into how ODE and SDE have distinct "winning regions", as illustrated in Figure 1 (b). The proof and claim for Theorem 1 should be reformulated such that even with $[t_{\min},t_{\max}] = [0,T]$, Theorem 1 explains how ODE and SDE have winning regions.
- Likewise, I think Theorem 2 also should be proven for the entire interval $[0,T]$, so we can directly compare the errors for SDEs, ODEs, and Restart.

**Weakness 2 : (Possibly) weak performance on the small NFE regime.**
- How does Restart perform in the small NFE regime (NFE $\leq 30$)? In Figure 3, the figure cuts off just before Restart and ODE intersect in the small NFE regime. This seems to contradict the claim that Restart combines the best of both ODE and SDE. Moreover, given the large size of SOTA diffusion models, it is crucial that diffusion samplers work well in the small NFE regime as well.
-  How does Restart compare to recent fast samplers such as [1], [2], [3] in the small NFE regime?

[1] DPM-Solver: A Fast ODE Solver for Diffusion Probabilistic Model Sampling in Around 10 Steps, NeurIPS, 2022.

[2] Fast Sampling of Diffusion Models with Exponential Integrator, ICLR, 2023.

[3] Denoising MCMC for Accelerating Diffusion-Based Generative Models, ICML, 2023.

**Questions:**

**Question 1** : How does Restart perform on FFHQ-1024 image generation (score function in https://github.com/yang-song/score_sde) compared to Improved SDE or ODE (Heun in EDM)?

**Limitations:**

Discussed in Section 6.

---

> ### Author Rebuttal · Authors · 2023-08-09
>
> Thank you for the detailed review and thoughtful feedback. Below we address specific questions.
>
> **Q1: For Theorem 1, if we set $[t_{min},t_{max}]=[0,T]$, the terms for contracted errors TV(p_{T}^{ODE_θ},pT) and TV(p_{T}^{SDE_θ},pT) vanish because p_{T}^{ODE_θ}, p_{T}^{SDE_θ}, and p_T are all identically Gaussian distributions. Then, we end up with terms depending on δ, ϵapprox, and $t_{max}−t_{min}$ only, so Theorem 1 does not provide any insight into how ODE and SDE have distinct "winning regions", as illustrated in Figure 1 (b). The proof and claim for Theorem 1 should be reformulated such that even with [t_{min},t_{max}]=[0,T], Theorem 1 explains how ODE and SDE have winning regions. Likewise, I think Theorem 2 also should be proven for the entire interval [0,T], so we can directly compare the errors for SDEs, ODEs, and Restart.**
>
> A:  Thank you for raising this question. First, we agree that if we set $[t_{min},t_{max}]=[0,T]$, Restart is the same as ODE and has no theoretical advantage. **However, we emphasize that the utility of restart is to reduce the accumulated error between $[t_{max}, T]$.** The interesting point is how this accumulated error is corrected during "Restart iterations". Therefore, setting $t_{min}=0$ and $t_{max}=T$ is a mis-application of Restart or SDE, as the accumulated error = 0 by definition.
>
> The main goal of Theorem 1 and 2 is to study how the already accumulated error changes using different samplers, and to understand their ability to self-correct the error by stochasticity. In essence, these theorems differentiate samplers based on their performance post-error accumulation. For example, by tracking the change of accumulated error, Theorem 1 shed light on the distinct "winning regions" of ODE and SDE: ODE samplers have smaller discretization error and hence excel at the small NFE regime. In contrast, SDE performs better in large NFE regime where the discretization error is negligible and its capacity to contract accumulated errors comes to the fore. We will clarify this point in our updated draft.
>
> **Q2: How does Restart perform in the small NFE regime (NFE ≤30)? How does Restart compare to recent fast samplers in the small NFE regime?**
>
> A: Thanks for the suggestions. We would like to highlight that the Restart sampler is compatible with fast ODE samplers, as they can be integrated in the deterministic backward process of Restart. As suggested by the reviewer, we compare Restart with DPM-Solver [3], a commonly-used fast ODE solver. In order to further accelerate Restart, we also use DPM-Solver in the main/Restart backward processes of Restart. We’ve included the FID versus NFE curves in Fig.1(a)  in the rebuttal PDF in the “Summary of Updates” comment above. The results show that the Restart consistently outperforms the DPM-Solver with a NFE ranging from 16 to 36. This demonstrates Restart's capability to excel over ODE samplers, even in the small NFE regime. Surprisingly, when paired with the DPM-Solver, Restart achieves a FID score of 2.11 on VP [1] when NFE=30, which is significantly lower than any previous numbers (even lower than the SDE sampler with a NFE $\ge 1000$ in [1]), and make VP model on par with the performance with more advanced models (such as EDM in [2]). We will include these results in our updated draft.
>
> **Q3: How does Restart perform on FFHQ-1024 image generation (score function in https://github.com/yang-song/score_sde) compared to Improved SDE or ODE (Heun in EDM)?**
>
> A: Thanks for the suggestion. We use the pre-trained FFHQ-1024 checkpoints in the code base pointed out by the reviewer (https://github.com/yang-song/score_sde), which is based on the Score-SDE-VE model with NCSN++ architecture. For stochastic baseline, we compare with the default Predictor-Corrector (PC) sampler [1] in the code base instead of Improved SDE, due to its need for time-consuming tuning of the hyper-parameters ($S_{tmax}, S_{tmin}, S_{noise}, S_{churn}$) on FFHQ. For ODE baseline, we compare with Heun as suggested by the reviewer. We set the NFE to 300 for all samplers.
>
>
> Since it’s prohibitively expensive to compute the FID score on this dataset, we qualitatively assess the visual quality from different samplers. We’ve included these images in Fig.2 in the rebuttal PDF in the “Summary of Updates” comment above. We observe that the Restart sampler produces notably superior image quality compared to other baselines. Heun sampler fails to generate clean images, and there is noticeable noise and artifacts of images generated by PC sampler compared to Restart. It indicates that the Restart sampler can successfully scale to 1024 resolution, and better balance the speed and quality in comparison to both ODE and SDE samplers. We will include the experiment in our revised version.
>
>
> *[1] Yang Song, Jascha Sohl-Dickstein, Diederik P. Kingma, Abhishek Kumar, Stefano Ermon, and Ben Poole. Score-based generative modeling through stochastic differential equations. ICLR 2021.*
>
> *​​[2] Tero Karras, Miika Aittala, Timo Aila, and Samuli Laine. Elucidating the Design Space of Diffusion-Based Generative Models. NeurIPS 2022.*
>
> *[3] Lu, Cheng, et al. "Dpm-solver: A fast ode solver for diffusion probabilistic model sampling in around 10 steps." Advances in Neural Information Processing Systems 35 (2022): 5775-5787*

---

> > ### Comment · Reviewer_kog8 · 2023-08-11
> > **Updated Score**
> >
> > I have raised the score by 1 point.
> >
> > I am not convinced by the result on FFHQ-1024, because I know it is possible to cherry-pick good samples on this dataset. I recommend the authors provide numerical results, e.g., FID, in the revised paper.

---

> > > ### Author Response · Authors · 2023-08-12
> > > **Thank you for your reply**
> > >
> > > We would like to thank the reviewer for the reply and suggestion. We would like to clarify that the samples provided in our rebuttal PDF were not selectively chosen. For a comprehensive comparison, extended samples for each method can be found in the `ffhq.pdf` via the anonymous link https://anonymous.4open.science/r/restart_rebuttal-3EE1/ffhq.pdf . These samples were generated using random seeds ranging from 1 to 9. A consistent observation is that the Restart sampler delivers markedly better image quality than other baselines throughout the batch. The Heun sampler struggles to produce clean images, while images generated by the PC sampler display noticeable noise and artifacts compared to Restart, using the Score-SDE-VE model.
> > >
> > > Given the extensive sampling time required for high-resolution data (FFHQ-1024) – approximately 38 seconds per image on a single A100 GPU – and the need for 50k generated images to achieve a reliable FID evaluation, we are not able to provide numerical results within the discussion period. We will to include these results in the revised version once they are available.

---

### Official Review · Reviewer_csQs · 2023-07-07

**Soundness:** 4 excellent
**Presentation:** 3 good
**Contribution:** 3 good
**Rating:** 6
**Confidence:** 4

**Summary:**

ODE-based samplers plateau in performance while SDE-based samplers deliver higher sample quality. The paper attributes this difference to discretization errors and accumulated errors. Based on these, the authors propose a sampling algorithm called Restart which alternates between the forward diffusion process and backward ODE.

**Strengths:**

1. The authors provide a theoretical explanation of the phenomenon that ODE samplers outperform SDE samplers in the small NFE regime but fall short in the large NFE regime.

2. The experimental results on image generation tasks validate the effectiveness of the method.

**Weaknesses:**

The proposed method relies on several hyperparameters (e.g. $S_{noise}, N_{restart}, i, K_i, t_{min, i}, t_{max, i}$), and the hyperparameters differ in different tasks. It would be hard to effectively tune these parameters in real application.


**Questions:**

Since the derivation of the Restart algorithm is motivated by the theoretical analysis of sampling error, is there a way to choose those hyperparameters based on the error analysis?

**Limitations:**

The author addressed the limitations of their work.

---

> ### Author Rebuttal · Authors · 2023-08-09
>
> Thank you for the detailed review and thoughtful feedback. Below we address specific questions.
>
> **Q1:  The proposed method relies on several hyperparameters (e.g. $N_{restart,i},K_i,t_{min},i,t_{max},i$), and the hyperparameters differ in different tasks. It would be hard to effectively tune these parameters in real applications. Since the derivation of the Restart algorithm is motivated by the theoretical analysis of sampling error, is there a way to choose those hyperparameters based on the error analysis?**
>
> A: Thanks for the question. Our choice of hyperparameter is partially motivated by theory. For instance, for a small $t_{min}$, we usually pick the parameter $t_{max} \approx B$, where $B$ is the radius of the dataset; this ensures that the contraction factor $\lambda$ in Theorem 4 is sufficiently large.
>
> Note that our theoretical results contain a number of Lipschitz constants, which are difficult to estimate in practice; thus our theoretical upper bounds help us decide high-level scaling but do not accurately prescribe a precise parameter choice. As another example, a larger accumulated error requires bigger Restart intervals and more Restart iterations, thus we use a larger $K$ and Restart interval for the weaker VP model [1] on CIFAR-10, compared to the EDM model [2].
>
> *[1] Yang Song, Jascha Sohl-Dickstein, Diederik P. Kingma, Abhishek Kumar, Stefano Ermon, and Ben Poole. Score-based generative modeling through stochastic differential equations. ICLR 2021.*
>
> *[2] Tero Karras, Miika Aittala, Timo Aila, and Samuli Laine. Elucidating the Design Space of Diffusion-Based Generative Models. NeurIPS 2022*
>
> ​​

---

> > ### Comment · Reviewer_csQs · 2023-08-18
> >
> > Thanks for your response.  I will keep my score based on the clarification. Please put the discussion in the final version.

---

### Official Review · Reviewer_kn2f · 2023-07-10

**Soundness:** 3 good
**Presentation:** 4 excellent
**Contribution:** 3 good
**Rating:** 7
**Confidence:** 4

**Summary:**

The papers propose a new sampling method for diffusion models, termed Restart Sampling. The authors first theoretically analyze the error propagation in diffusion models for stochastic and deterministic samplers under Wasserstein-1 distance and show that ODE samplers have a lower-discretization error but SDE samplers contract the initial distribution error as we run more steps. This agrees with the intuition and the experimental findings that support that ODE samplers are better for low NFEs but their performance flattens for more NFEs. Based on this analysis, the authors propose a method that tries to achieve the best of both worlds: it contracts the initial error with more steps and achieves the same discretization error as the ODE samplers. The implementation of the new method is very straightforward: one runs the ODE sampler and every K steps reverts back to some prior diffusion time using the forward model.

**Strengths:**

The authors study a relevant problem in diffusion models. The proposed solution is simple, effective and novel. The theoretical results motivate the method and show clearly the differences between the SDE and the ODE samplers. The presentation of the paper is excellent. The authors show many experimental results, starting from toy models and going all the way to state-of-the-art text-to-image diffusion models. I think the paper and the method is of interest to the community and the audience of NeurIPS.

**Weaknesses:**

The error propagation of diffusion models has been studied before. The results that I am aware of are from the papers "Sampling is as easy as learning the score", "Restoration-degradation beyond linear diffusions: A non-asymptotic analysis for DDIM-type samplers" and "The probability flow ODE is provably fast". The first studies the error propagation of the SDE sampling method and the latter two the propagation of errors for deterministic samplers. It would be beneficial to compare with these works, highlight potential differences in the approach and the final results, etc.

Also, apart from the Stochastic and the ODE samplers, there is a whole family of samplers that satisfy the same Fokker-Planck equations and hence give the same marginals, e.g. see the work "Fast Sampling of Diffusion Models with Exponential Integrator" and also some of the samplers used in the "Elucidating the Design Space of Diffusion-Based Generative Models" (EDM) paper. It would be interesting to compare theoretically and experimentally to these samplers.

Another concern I have is that the evaluation is only as thorough as it should have been and it is only done for relative high NFEs. Since evaluating the performance of a trained model is relatively easy, I would expect a more thorough benchmarking. If the performance of Restart sampler breaks for low NFEs, it is useful to know it and acknowledge it in the paper.



**Questions:**

I think it would be useful to know:

* how the performance is for lower NFEs, e.g. 20 sampling steps?
* the behavior of the sampler across more datasets and NFEs. For example, it would be useful to include more comparisons with the EDM paper (and the references therein) for different datasets.
* what are the differences in the theoretical analysis compared to prior results known for error-propagation in diffusion models.

**Limitations:**

The authors adequately addressed the limitations of their work.

---

> ### Author Rebuttal · Authors · 2023-08-09
>
> Thank you for the detailed review and thoughtful feedback. Below we address specific questions.
>
>
> **Q1: The error propagation of diffusion models has been studied before ... What are the differences in the theoretical analysis compared to prior results known for error-propagation in diffusion models.**
>
>
> A: Thank you for pointing out these related works. We will add a more detailed comparison to the draft. Briefly, our result differs from the above-mentioned papers in the following key way: [1,2] aim to control the discretization error going from $T \to 0$ during the backward process.  [1] does this using Girsanov’s theorem, and [2] does this using a more involved analysis as one cannot apply Girsanov in the absence of diffusion noise. In contrast, our analysis focuses on how Restart iterations over the sub-interval $[t_{min} , t_{max}]$ reduces the accumulated discretization error from $[t_{max},T]$. There are some similarities with [3], notably they also have a corrector stage where they run overdamped LMC. One important difference from [3] is that our Restart iterations are not interleaved with ODE integration.
>
> A remarkable flexibility of our analysis is that **we need to make very little assumption about the accumulated error up to $t_{max}$**. Consequently, one can adopt a wide range of integration techniques from [t_{max}, T], and apply Restart at the end (over $[t_{max}, t_{min}]$) to reduce the accumulated integration error between $[t_{max}, T]$. In contrast, the analysis in [1,2,3] is focused on specific ODE/SDE integration algorithms.
>
> We also note that [3] is a concurrent work and appeared on Arxiv after our submission to NeurIPS. We will include these discussions in our revised version.
>
>
> **Q2: Apart from the Stochastic and the ODE samplers, there is a whole family of samplers that satisfy the same Fokker-Planck equations and hence give the same marginals, e.g. see the work "Fast Sampling of Diffusion Models with Exponential Integrator"(DEIS) and also some of the samplers used in the EDM paper. It would be interesting to compare theoretically and experimentally to these samplers.**
>
> A: We would like to note that the samplers highlighted by the reviewer are encompassed within the either stochastic or ODE samplers we've discussed, and we've already made some comparisons in our paper. Specifically, we've compared the ODE sampler (Heun) in EDM and the recommended stochastic sampler in EDM (Improved SDE). Please refer to Fig.3, Table.1, Fig.8, and Fig.10 in the draft for details.
> We appreciate the suggestion to compare faster ODEs with exponential integrators. We've incorporated DDIM in the draft, which serves as the first-order counterpart to DEIS. Our new empirical results demonstrate that Restart can improve over DPM-Solver, which is also a higher-order ODE solver that utilizes exponential integrators. The subsequent question's response provides more detailed results.
>
> **Q3: Another concern I have is that the evaluation is only as thorough as it should have been and it is only done for relatively high NFEs. How is the performance for lower NFEs, e.g. 20 sampling steps?**
>
> A: Thanks for the suggestion. We employ the recommended Heun’s 2nd order ODE sampler [4] in the main backward process of Restart in our draft. Since Heun’s 2nd order ODE is not targeting the low NFE regime, we did not emphasize the performance of such regime. To validate the effectiveness of the Restart sampler in the low NFE regime, we use the faster ODE sampler in [5] (DPM-Solver) in the backward processes of Restart. We’ve included the FID versus NFE curves in Fig.1(a)  in the rebuttal PDF in the “Summary of Updates” comment above. The results show that the Restart consistently outperforms the DPM-Solver with an NFE ranging from 16 to 36. This demonstrates Restart's capability to excel over ODE samplers, even in the small NFE regime. We will include these results in our updated draft.
>
>
> **Q4: The behavior of the sampler across more datasets and NFEs. For example, it would be useful to include more comparisons with the EDM paper (and the references therein) for different datasets.**
>
> A: Thanks for the suggestions. Following the suggestion, we evaluated the Restart sampler on smaller NFEs, particularly when paired with the DPM-Solver, as previously discussed. We've also highlighted the efficiency of the Restart sampler on the FFHQ-1024 dataset, as depicted in Fig.2 of the rebuttal PDF. We will include more datasets in the updated draft.
>
> *[1] Sampling is as easy as learning the score, Chen et al., ICLR 2023*
>
> *[2] Restoration-degradation beyond linear diffusions: A non-asymptotic analysis for DDIM-type samplers, Chen et al., ICML 2023*
>
> *[3] The probability flow ODE is provably fast, Chen et al., arXiv 2305.11798*
>
> *​​[4] Elucidating the Design Space of Diffusion-Based Generative Models, Karras et al., NeurIPS 2022.*
>
> *[5]  Dpm-solver: A fast ode solver for diffusion probabilistic model sampling in around 10 steps, Lu et al. NeurIPS 2022.*

---

> > ### Comment · Reviewer_kn2f · 2023-08-12
> > **Respone to Rebuttal**
> >
> > Thank you for your rebuttal. Please incorporate these discussions in the camera-ready version of your work, if possible. I will increase my score to 7.

---

### Official Review · Reviewer_YGP7 · 2023-07-11

**Soundness:** 3 good
**Presentation:** 3 good
**Contribution:** 2 fair
**Rating:** 5
**Confidence:** 3

**Summary:**

By analyzing of the trade-off between good sample quality and sampling time of both ODE and SDE-based generative models, a restart sampling strategy is proposed by this paper to combine the advantages of ODE and SDE sampling methods. The author proves two theorems that estimate the upper bound on the total error measured by the Wasserstein distance between generated and data distributions of ODE, SDE, and restart sampling methods respectively. It is illustrated that the total error can be decomposed into two parts: additional sampling error generated by discretization error and contracted error generated by the accumulated total error from previous sampling steps. Moreover, it is proved that ODE-based samplers have smaller additional sampling errors and SDE-based samplers have smaller contracted errors. So Comparing the three upper bounds on the total error, it can be proved theoretically that restart sampling yields a smaller total error because its additional sampling error and contracted error are both small. Finally, a range of experiments are done by authors which shows empirically that: 1. The total error of the restart sampler is indeed smaller than that of others. 2. Restart sampler surpasses previous SDE and ODE samplers in both speed and accuracy. 3. Restart sampler better balances text-image alignment/visual quality versus diversity than previous samplers.

**Strengths:**

1. This paper is written with meaningful motivation and a clear structure. The restart sampling method proposed by this paper is innovative, simple, and effective.
2. The reasonableness and effectiveness of the resampling method are proved both theoretically and experimentally.
3. The upper bounds on the total error of the three sampling methods give us an intuitive understanding of the advantages and disadvantages of the three sampling methods.

**Weaknesses:**

1. The effectiveness of restart sampling method on high-resolution image synthesis is not confirmed. For example, a comparison of sampling speed and accuracy on ImageNet 128*128, 512*512 should be added.
2. The paper experiments on the sensitivity analysis of the number of restart iterations K, but there is no experiment on the sensitivity of another hyperparameter: the position and length of the restart interval.
3. As pointed out in the paper, the contracted error further diminishes exponentially with the number of repetitions K though the additional error increases linearly with K. Figure 4 illustrates this trade-off phenomenon, too. So it may be hard to find a suitable K to make the restart algorithm work for different datasets or tasks globally, making it difficult to apply.

**Questions:**

1. Although the sampling step decreases by applying the restart strategy, the repetiness of ODE solver in the restart interval seems time-consuming. Moreover, does NFE computed in the restart interval added to the total NFE in Figure 3?
2. Why does the total error remain the same using the ODE sampler when NFE>20? What are the results of three samplers with NFE<20? Is it consistent with the trend in Figure 1(b)？
3. Why not compare to DPM-solver? Moreover, why not apply restart to DPM-solver to see if there are consistent benefits?

---

> ### Author Rebuttal · Authors · 2023-08-09
>
> Thank you for the detailed review and thoughtful feedback. Below we address specific questions.
>
> **Q1: The effectiveness of Restart sampling method on high-resolution image synthesis is not confirmed. A comparison of sampling speed and accuracy on ImageNet-128/-512 should be added.**
>
> A: Thanks for the suggestion. We agree with the reviewer that including the suggested datasets can strengthen our experimental results. However, we would like to note that we have already conducted experiments at a resolution of 512x512, on the Stable Diffusion model. In particular, Restart demonstrates a superior FID and CLIP/aesthetic score trade-off, underscoring its scalability to higher-resolution images. As suggested by reviewer kog8, we've also presented a qualitative demonstration of Restart's effectiveness on FFHQ-1024 (please refer to Fig.2 in the rebuttal PDF).
>
> **Q2: There is no experiment on the sensitivity of another hyperparameter: the position and length of the restart interval.**
>
> A: Thank you for the suggestion. In Fig.9 of our paper, we illustrated the sensitivity of $t_{min}$ while keeping $t_{max}$ constant. We additionally add more sensitivity analysis of both the position and length of the restart interval on CIFAR-10. We only implement one Restart interval in all the experiments. We include the results in Fig.3 in the rebuttal PDF.
>
> For sensitivity to Restart length $t_{max}-t_{min}$, we fix $t_{min}$ at 0.06 for VP and 0.14 for EDM. In theory, a longer interval enhances contraction but may add more additional sampling errors. Again, the balance between these factors results in a V-shaped trend in our plots (Fig.3(a)). In practice, selecting $t_{max}$ close to the dataset's radius usually ensures effective mixing when $t_{min}$ is small.
>
> For sensitivity to $t_{min}$, Fig.3(b) shows that a moderately small $t_{min}$ minimizes the approximation error post-restart on CIFAR-10. However, the contraction effect diminishes as $t_{max} - t_{min}$ diminishes.
>
> **Q3: It may be hard to find a suitable $K$ to make the restart algorithm work for different datasets or tasks globally, making it difficult to apply.**
>
> A: Thanks for pointing this out. We agree that fully optimizing the hyper-parameter $K$ would be challenging. However, in general, the quality of generated images initially improves and later worsens with an increasing $K$. This trend makes it feasible to identify an appropriate $K$ value through a straightforward binary search.
>
> In our experiments, we've found that choosing a suitable value for $K$ that leads to improved performance is relatively easy.  For example, introducing a Restart interval with $K=2$ at small time consistently outperforms the baselines for all the datasets given the same NFE. In addition, as a heuristic, one could set $K$ to reasonable values by following the recipe that at a smaller time $t$, a larger $K$ is necessary to contract more accumulated errors. Nevertheless, we acknowledge that determining the optimal $K$ for different Restart intervals could be intricate. We will delve deeper into this in future studies.
>
> **Q4: Although the sampling step decreases by applying the restart strategy, the repetiness of ODE solver in the restart interval seems time-consuming. Moreover, does NFE computed in the restart interval added to the total NFE in Figure 3?**
>
> A: Yes, the total NFE reported in the paper includes both the NFE in Restart intervals as well as the NFE in the main backward process. Even though each Restart interval involves several function evaluations, the overall FID-NFE trade-off in Restart remains superior to previous methods. This is primarily because Restart allows for a reduced NFE during the main backward process.
>
> **Q5: Why does the total error remain the same using the ODE sampler when NFE>20 (Fig.2)? What are the results of three samplers with NFE<20? Is it consistent with the trend in Figure 1(b)**
>
> A: Fig.2 plots the Pareto frontier of total Error versus NFE. For ODE, we have conducted experiments with NFE 20,40,80,160,320,640, with total error 0.89, 0.90, 0.90, 0.90, 0.90, 0.90 respectively. This reveals that the Pareto front stabilizes at an NFE of 20 with an error of 0.89. A larger NFE didn’t reduce the error since the discretization error is already small.
>
> We conducted additional experiments (please see Fig.1(a) in the rebuttal PDF) and verified that the trend is consistent when NFE is less than 20. We will include the result in the updated draft.
>
> **Q6: Why not compare to DPM-solver? Why not apply Restart to DPM-solver to see if there are consistent benefits?**
>
> A: Thanks for the suggestions. As suggested by the reviewer, we compare Restart with DPM-Solver. In order to further accelerate Restart, we also use DPM-Solver in the main/Restart backward processes of Restart. We’ve included the FID versus NFE curves in Fig.1(a)  in the rebuttal PDF in the “Summary of Updates” comment above. The results show that the Restart consistently outperforms the DPM-Solver with an NFE ranging from 16 to 36. This demonstrates Restart's capability to excel over ODE samplers, even in the small NFE regime. It also suggests that Restart can consistently improve other ODE samplers, not limited to the DDIM, Heun in the paper. Surprisingly, when paired with the DPM-Solver, Restart achieves an FID score of 2.11 on VP [1] when NFE=30, which is significantly lower than any previous numbers (even lower than the SDE sampler with an NFE $\ge 1000$ in [1]), and make VP model on par with the performance with more advanced models (such as EDM in [2]). We will include these results in our updated draft.
>
> *[1]  Yang Song, Jascha Sohl-Dickstein, Diederik P. Kingma, Abhishek Kumar, Stefano Ermon, and Ben Poole. Score-based generative modeling through stochastic differential equations. ICLR 2021.*
>
> *​​[2] Tero Karras, Miika Aittala, Timo Aila, and Samuli Laine. Elucidating the Design Space of Diffusion-Based Generative Models. NeurIPS 2022.*

---

> > ### Comment · Reviewer_YGP7 · 2023-08-13
> > **Thanks**
> >
> > Thanks for the rebuttal. I like the direct comparison with dpm-solver. I suggest the authors carefully update the paper in the final revision to make clear the points raised in the review. I have increased my score.

---

### Official Review · Reviewer_HtQK · 2023-07-13

**Soundness:** 3 good
**Presentation:** 4 excellent
**Contribution:** 2 fair
**Rating:** 6
**Confidence:** 3

**Summary:**

The paper proposed a sampler that balances sampling speed and quality by adding noises and restarting the process. They provide theoretical analysis to show a better upper bound of this method compared to original ODE and SDE samplers. Experiments are done to verify their claims.

**Strengths:**

1. Authors identify the main cause of different performances of SDE and ODE in different regimes. And by taking advantage of the contraction by adding noises, they balance the speed and quality of the sampler.

2. The theoretical analysis is clear and well-written.

3. The experiments are thorough with a good explanation of the choices of hyperparameters.

4. The experiments show good results for the proposed method.

**Weaknesses:**

See questions.

**Questions:**

I wonder what the wall clock time looks like between restart, sde, and ode.

**Limitations:**

yes

---

> ### Author Rebuttal · Authors · 2023-08-09
>
> Thank you for the detailed review and thoughtful feedback. Below we address specific questions.
>
> **Q1: I wonder what the wall clock time looks like between Restart, SDE, and ODE.**
>
> A: Thank you for the question. The wall clock time during sampling is approximately proportional to the NFE (number of function evaluations), which is reported in the paper. This is because the primary computational bottleneck during sampling lies in evaluating the neural networks. Any other overhead, in comparison, is negligible. Therefore, when comparing Restart, SDE, and ODE, their wall clock time will closely align with their respective NFEs.

---

> ### Comment · Area_Chair_tisB · 2023-08-20
> **Reminder from AC**
>
> Dear reviewer,
>
> The author-reviewer discussion period ends in 2 days. Please review the authors' rebuttal and engage with them if you have additional questions or feedback. Your input during the discussion period is valued and helps improve the paper.
>
> Thanks, Area Chair

---

### Author Rebuttal · Authors · 2023-08-10

# Summary of Updates

We would like to thank all reviewers for their constructive feedback. We have revised our draft according to all the valuable comments. Below we summarize updates in the revised version. We also include all the new figures in the PDF files attached.

## 1. More experiments

In response to Reviewer YGP7, kn2f and kog8, we have compared Restart with DPM-Solver and incorporated experiments in the low NFE regime. We’ve included the FID versus NFE curves in Fig.1(a)  in the rebuttal PDF. The results show that the Restart consistently outperforms the DPM-Solver with an NFE ranging from 16 to 36. As recommended by Reviewer kog8, we qualitatively validate the effectiveness of Restart on FFHQ-1024 dataset (Fig.2 in the rebuttal PDF). Additionally, we have included smaller NFEs in the study of total error versus NFE (Fig.1(b) in the rebuttal PDF), and additional sensitivity analysis of hyperparameters (Fig.3 in the rebuttal PDF), as suggested by Reviewer YGP7.

## 2. Discussion on related works / Clarification on theorems

As suggested by Reviewer k2nf, we’ve added the discussion of the difference between Restart and prior or concurrent works. In response to Reviewer kog8, we have also provided clarifications on our theorems, emphasizing their objectives in examining the behavior of various samplers with accumulated errors.

---

### Decision · Program_Chairs · 2023-09-21

**Decision:**

Accept (poster)

**Comment:**

Based on my assessment of the reviews, I recommend accepting this paper. The work proposes a novel sampling algorithm for diffusion models that aims to balance speed and quality. The core idea of adding noise and restarting sampling shows promise in improving over prior methods.

The reviewers rate the paper mostly as Weak Accept or Accept, with multiple increasing scores after the rebuttal. They find the theory and experiments to be technically solid. The qualitative gains on high-resolution image synthesis and comparisons to recent fast samplers address initial concerns. The discussion helped clarify ambiguities in the theorems.

While further experiments on scalability would be beneficial, the paper makes a meaningful contribution in improving sampling for generative diffusion models. The combination of new algorithm design, analysis, and empirical gains over strong baselines highlights the potential of this approach. I recommend acceptance to share this work with the broader community through publication. Additional experiments can be incorporated into the camera ready version.